# Fiber Optic Sensors for Vital Signs Monitoring. A Review of Its Practicality in the Health Field

**DOI:** 10.3390/bios11020058

**Published:** 2021-02-23

**Authors:** Christian Perezcampos Mayoral, Jaime Gutiérrez Gutiérrez, José Luis Cano Pérez, Marciano Vargas Treviño, Itandehui Belem Gallegos Velasco, Pedro António Hernández Cruz, Rafael Torres Rosas, Lorenzo Tepech Carrillo, Judith Arnaud Ríos, Edmundo López Apreza, Roberto Rojas Laguna

**Affiliations:** 1Doctorado en Biociencias, Facultad de Medicina y Cirugía, Universidad Autónoma “Benito Juárez” de Oaxaca, Ex Hacienda de Aguilera S/N, Calz. San Felipe del Agua, 68050 Oaxaca de Juárez, Mexico; jcano@uabjo.mx; 2Escuela de Sistemas Biológicos e Innovación Tecnológica, Universidad Autónoma “Benito Juárez” de Oaxaca (SBIT-UABJO), Av. Universidad S/N, Ex-Hacienda 5 Señores, 68120 Oaxaca de Juárez, Mexico; mvargas.cat@uabjo.mx (M.V.T.); ltepech@uabjo.mx (L.T.C.); elopez.cat@uabjo.mx (E.L.A.); 3Centro de Investigación Facultad de Medicina UNAM-UABJO, Facultad de Medicina y Cirugía, Universidad Autónoma “Benito Juárez” de Oaxaca, Ex Hacienda de Aguilera S/N, Calz. San Felipe del Agua, 68050 Oaxaca de Juárez, Mexico; itanbel@hotmail.com (I.B.G.V.); phernandez.cat@uabjo.mx (P.A.H.C.); 4Facultad de Odontología, Universidad Autónoma “Benito Juárez” de Oaxaca, Av. Universidad S/N, Ex-Hacienda 5 Señores, 68120 Oaxaca de Juárez, Mexico; rtorres.cat@uabjo.mx; 5Doctorado en Ciencias en Desarrollo Regional y Tecnológico, Tecnológico Nacional de México Campus Oaxaca, Avenida Ing. Víctor Bravo Ahuja No. 125 Esquina Calzada Tecnológico, 68030 Oaxaca de Juárez, Mexico; judith.ar23@gmail.com; 6Departamento de Electrónica, División de Ingeniería, Universidad de Guanajuato, Carretera Salamanca-Valle de Santiago km 3.5 + 1.8, Comunidad de Palo Blanco, 36885 Salamanca, Mexico; rlaguna@ugto.mx

**Keywords:** fiber optic sensor, vital signs, biosensor, human body, body temperature, heart rate, respiratory rate, blood pressure

## Abstract

Vital signs not only reflect essential functions of the human body but also symptoms of a more serious problem within the anatomy; they are well used for physical monitoring, caloric expenditure, and performance before a possible symptom of a massive failure—a great variety of possibilities that together form a first line of basic diagnosis and follow-up on the health and general condition of a person. This review includes a brief theory about fiber optic sensors’ operation and summarizes many research works carried out with them in which their operation and effectiveness are promoted to register some vital sign(s) as a possibility for their use in the medical, health care, and life support fields. The review presents methods and techniques to improve sensitivity in monitoring vital signs, such as the use of doping agents or coatings for optical fiber (OF) that provide stability and resistance to the external factors from which they must be protected in in vivo situations. It has been observed that most of these sensors work with single-mode optical fibers (SMF) in a spectral range of 1550 nm, while only some work in the visible spectrum (Vis); the vast majority, operate through fiber Bragg gratings (FBG), long-period fiber gratings (LPFG), and interferometers. These sensors have brought great advances to the measurement of vital signs, especially with regard to respiratory rate; however, many express the possibility of monitoring other vital signs through mathematical calculations, algorithms, or auxiliary devices. Their advantages due to miniaturization, immunity to electromagnetic interference, and the absence of a power source makes them truly desirable for everyday use at all times.

## 1. Introduction

The penetration of fiber optic sensors in the medical or health care market is very low or almost null in home devices due to their high cost or too many regulations, which hinder their entry; however, there have been small advances as these sensors are used in high precision surgery or in magnetic resonance imaging; their use is increasing in various technologies or ancillary equipment, so it has advanced one step at a time [1].

Since the 1980s [2], optical fiber sensors have been used for real-time pressure measurement of tendons [3] and thereafter in a wide variety of possible medical applications as pressure pads in contact with the skin [4], the cardiovascular system, or even invasive sensors during urodynamic analysis [5]. Many researchers have focused their studies on monitoring and detecting vital signs through fiber optic sensors that are located in some medium that is in contact with the skin [6] and with the capacity to be composed of two or more sensors on the same fiber [7]. Such applications may be placed in textile vestments [8,9] that are very novel in their shape or that have an ingenious way of installing them, while at the same time providing protection to the optical fiber [10]. Because vital signs are the first way to test a person’s health and stability, heart pulse monitoring, being a well-studied field [11], is one area that lends itself to continuing ways of innovating and developing improvements in measurement devices and equipment [12,13].

Fiber optic sensors have been used in many applications for the measurement of chemical parameters, liquid flow and levels, and gas detection. Nonetheless, they have been mainly used in the electrical and mechanical fields due to the great advantages that have been attributed to them, such as immunity to electromagnetic interference, apart from the small size of the fiber that makes them perfect for the development of lightweight and mechanically robust sensors [14]. However, in the medical field they have not yet gained wide acceptance since conventional sensors have also made great advances, giving them good user characteristics such as reliability, maintenance and support, and technological integration [15]. There is a great need for a simple system for measuring vital signs for home health monitoring.

## 2. Sensor Principles

In the next analysis, we considered various studies based on the premise of dividing sensor measurement of four vital signs that are: body temperature, respiration or breathing rate (BR), pulse or heart rate (HR), and blood pressure. Here, we illustrate the main characteristics of each sensor, how measurements can be obtained from each of them, and the peculiarities of these sensors. Furthermore, we examine the common characteristics among other methods of vital signs monitoring. In addition, the basic technologies used by each sensor are briefly presented, including common characteristics, features, and qualitative techniques for their characterization. The tables present the most complete articles in terms of the parameters that we took into consideration for this review, such as the optical fiber (OF) type and wavelength at which the sensor operates, type of technology used, operating ranges, sensitivity, and medium of contact with the test subject.

### 2.1. Fiber Bragg Gratings (FBGs)

FBGs have been widely used in various sectors of industry to measure deformations [16]. They are periodic diffraction modifications printed on the core of an OF, behaving as selective filters that reflect only the components of the spectrum of a packet propagating in the core according to the Bragg relation
(1)$$\lambda =2n\Lambda_{L}$$
where the wavelength is represented as *λ*, *n* is the effective refractive index of the core mode, and *Ʌ_L_* the spatial period of the refractive index modulation. If the FBG is strained along the fiber axis, it changes as does the Bragg wavelength as it moves, resulting in a measurement of strain or temperature [17]; this allows for high resolution, accuracy, and dynamic range suitable for applications such as biomedicine and can be calculated according to the equation:(2)$$\Delta \lambda_{\beta }=2\left(\Lambda \frac{\partial n_{eff}}{\partial \varepsilon }+n_{eff}\frac{\partial \Lambda }{\partial \varepsilon }\right)\Delta \varepsilon_{z}+2\left(\Lambda \frac{\partial n_{eff}}{\partial T}+n_{eff}\frac{\partial \Lambda }{\partial T}\right)\Delta T$$

Here, the first term represents the dependence of the pressure on the Bragg wavelength and the second term represents the effect of temperature on the same parameter. It is shown that an FBG can be used as a sensor by observing the light reflected through the FBG for the longitudinal mechanical strain ***ε_z_***, and the temperature *T*. For dynamic loading, the heat input can be neglected and the sensitivity of the FBG for ***ε_z_***, is expressed according to the equation:(3)$$\frac{\Delta \lambda_{\beta }}{\lambda_{\beta }}=\left\{\left(1-\left(\frac{n^{2}}{2}\right)\right)\left[p_{12}-v\left(p_{11}+p_{12}\right)\right]\right\}\varepsilon_{z}$$
where *v* and *p_ij_* are mechanical properties of the coating; therefore, Equation (3) can be simplified as:(4)$$\frac{\Delta \lambda_{\beta }}{\lambda_{\beta }}=\left(1-p_{e}\right)\varepsilon_{z}$$

When *p_e_* is the optical pressure coefficient of the OF, the Bragg wavelength is directly proportional to the length of the grating because the deformation of the Bragg wavelength can be observed, and the induced deformation can be controlled [18]. Figure 1 shows how an FBG operates, where the core of the fiber is modified through periodic inscriptions that will act as a filter for a certain wavelength of light. Of the transmitted light, the grating will only return the reflected light.

#### 2.1.1. Long-Period Fiber Gratings (LPFGs)

Another common technique to change the fiber nanostructure is through a mechanically induced LPFG (MLPFG). Such a sensor has been shown to perform well as a mode-coupled device, pressure sensor, and filter [20]. Its working principle turns out to be quite as simple as pressing an OF between a ribbed plate and a flat plate, where the ribbed plate has to carry inscribed periods ranging in size from 100 µm to 1000 µm. Figure 2 shows how gratings with these periods inscribed in the core only let through certain light at some wavelength; its feature is that it promotes coupling between *LP*_01_ core mode and jointly propagating antisymmetric *LP*_1*m*_ (*m* > 0) and OF cladding modes, resulting in transmission spectra containing a series of attenuation notches for discrete wavelengths, which satisfy the phase matching condition
(5)$$\lambda_{res}^{n}=\left(n_{effcore}-n_{effclad\left(n\right)}\right)\Lambda$$
where *n_effcore_* and *n_effclad(n)_* are the effective refractive indices of the core and the nth mode of the cladding at the resonant wavelength $\lambda_{res}^{n}$, and *Ʌ* is the period of the grating [21].

However all fiber optic-based devices are considered immune to electromagnetic interference [22]. Fiber gratings can be affected in their wavelength by deformation and temperature. This same cross sensitivity to pressure and temperature has proven to be a major obstacle to the development of practical applications, but, in recent years, interferometer- based sensors to measure pressure have become a major object of analysis and study, becoming quite popular for different laboratory tests [23]. Among those that stand out are interferometers that use the interference of light waves to be able to accurately measure wavelengths of light themselves, very small distances, and certain optical phenomena through two light beams that travel different optical paths, determined by a system of mirrors or plates that in the end converge to form an interference pattern. The Sagnac, Mach–Zehnder interferometer (MZI), Fabry–Pérot interferometer (FPI), and Michelson interferometer stand out as OF sensors based on multimode interference, offering advantages for their low production cost, high sensitivity, and robustness compared to other fiber sensors [24].

**Figure 2 biosensors-11-00058-f002:**
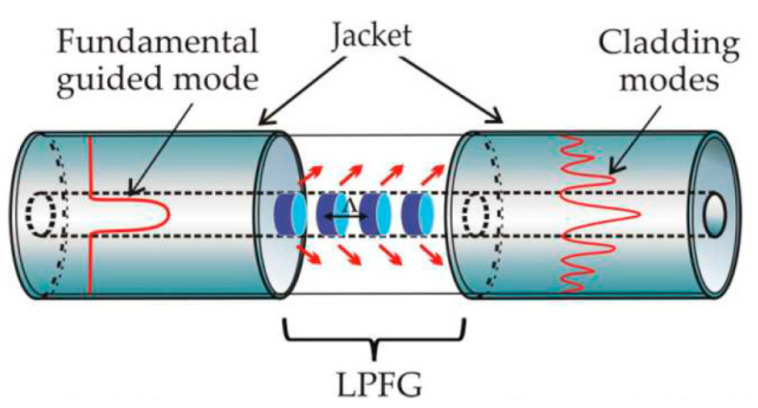
Schematic of a long-period fiber grating (LPFG) and the mode of propagation of light through the core, where gratings separated by defined periods perform the filtering function [25].

These gratings are a periodic modulation induced by mechanical action or through some chemical or laser to generate such periods by modifying the refractive index of the fiber core. The effect of these LPFGs is the induction of a series of attenuation bands in the transmission shape of the fiber that produce wavelengths at which the LPFGs induce a phase coupling between the guided and cladding modes. The study of LPFG attenuation bands has led to many potential applications, including the medical and sensing area, since the wavelengths of the attenuation bands are sensitive to the strain, temperature, and refractive index of the surrounding medium [26].

#### 2.1.2. Mechanical Induction and Fiber Optic Sensors

Mechanical induction through microbend is one of the earliest forms of sensor characterization; the losses caused by these microbends have been a major problem for fiber cable designers, but it has been these same losses that have helped many sensor designers to adapt to the effects caused by the bends to measure many physical parameters and variables, such as temperature and pressure. With some outstanding performance characteristics, they have been successful for some commercial applications. Additionally, there have been many advances in understanding microbend sensors and investigating how to increase the dynamic range and improve sensitivity to the measurement parameter of interest while reducing sensitivity to unwanted variables. Among the great advantages of microbend sensors are mechanical and optical efficiency that allow for low part count and low cost and easy mechanical assembly that does not require fiber optic splices with other components and therefore avoids the problems of thermal expansion difference. The Figure 3, shows an example of how the fiber (yellow) is crushed as force is exerted through the plates when they are brought together, an attempt to inscribe the periods of the plate ribs to the OF. These are fail-safe sensors, as they produce a calibrated output signal in its correct function or when fail completely, immediately going to a no-light output state [27].

### 2.2. Optical Fiber Interferometers

Continuing with the operation principles, we can analyze the utility of the interferometers as devices whose operation is in effect caused by the interference of two beams of light, which propagate by different optical paths through one or more optical fibers [29]. It is the form of operation in which a light beam is separated or is grouped that is required for these modules to work with optical fibers; we can basically emphasize four types of interferometers whose configurations are those of Fabry–Pérot, Mach–Zehnder, Michelson, and Sagnac—all of which have been widely used and demonstrated [30]. The main characteristics and differences among them in their application as optical fiber sensors will be described.

Used for extremely precise measurements in very small distances, its design also is known as “etalon”. Most conventional optical interferometers have their equivalent in OF. In these interferometers the light is guided by the fibers and the splitter plates are replaced by directional couplers for OF. The mirrors in some cases can be replaced by the fiber ends themselves properly terminated. Figure 4, shows the shape of an optical arrangement of an FPI, where the light beam from the source is reflected multiple times (causing interference) through the parallel surfaces, and the reflected light is sent to the detector by an optical lens.

In optical fibers, one way that it is possible to be created is through a ring (gyroscope) that works as a guide of a wave; this is possible if the optical fiber conserves a defined front of affluent wave and if the effectiveness in the connection of the light in the fiber is not very small [31].

**Figure 4 biosensors-11-00058-f004:**
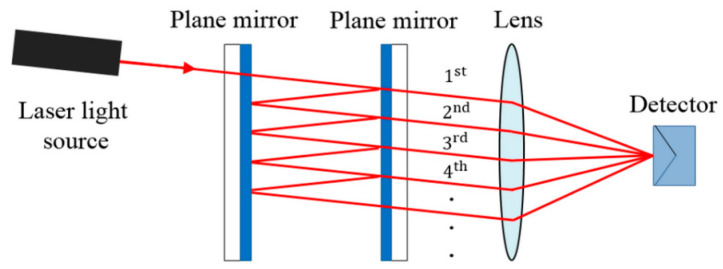
Sample of the operation of the Fabry–Pérot interferometer (FPI) [32].

#### 2.2.1. Optical Fiber Mach–Zehnder Interferometer

Its structure consists of two couplers that divide the optical power of entrance in two equal parts; thus, the light travels through two different ways—one way operating as a reference while the other is used for sensing, it’s desirable that OF have the same length so that the interference pattern is constructive, and the phases are equal. In Figure 5a it is possible to observe that since the beam of incident light is divided in two arms by means of a coupler later to recombine itself by means of a second coupler, later to recombine itself by means of a second coupler, the recombined beam of light has the interference component depending on the path optical between the arms. When this happens, for the detection, the reference arm stays isolated from any external variation, and another arm (the sensing region) is exposed to the changes. The changes in the arm (the sensing region) are due the mode propagation of the beam, that is sensitive to the refractive index changes of the materials of the sensor. This refractive index changes, are what originate the difference between paths optical, with which it is possible to later analyze the variation in the interference signal with photodetector [33].

#### 2.2.2. Optical Fiber Michelson Interferometer

In the optical fiber sensor that is presented in Figure 5b, the coupler splits the beam of light by two different optical methods, where the light reflected by the mirrors recombines them by means of the coupler, giving origin to an interference pattern that arrives for analysis at the photodetector. It can be said that this interferometer’s method of operation is very similar to the MZI; in fact, a Michelson is like half of an MZI as far as its configuration, since it uses a single coupler and a photodetector. Basically, the interference takes place between the beams of both arms, but each one of the light beams is reflected at the end of each arm in a Michelson interferometer. The main difference between these two interferometers are their mirrors —but their manufacture and principles of operation are almost equal. In this device, Faraday rotator mirrors are used to maintain the polarization of the separated beams in the fiber [34].

#### 2.2.3. Optical Fiber Sagnac Interferometer

In this type of interferometer, which is seen in Figure 5c, the light beam enters the coupler by one of the entrance fibers, as it is possible to observe in the figure, giving rise to the light that divides in two beams with the same intensity, but with each one of them moving in opposing directions through the optical fiber, which in the end is going to be on the side of the configuration of the Sagnac—the one that goes toward the photodetector. This phenomenon is known as detection of rotation, which is obtained by placing the device on a spin table, which is when these turns happen and the lines of the interference pattern are moved [35].

#### 2.2.4. Optical Fiber Fabry–Pérot Interferometer

This interferometer is considered the simplest since its single configuration needs only a circulator and a photodetector. For this interferometer, the interference plays a fundamental role since it is originated by means of a cavity in some of the ends of the optical fiber, as is in Figure 5d, where a light beam that is originated at the source travels through the fiber toward the circulator. When the light bean arriving at the cavity, the interference due to the light beam is reflected into cavity, to returns toward the photodetector. The cavity of a Fabry–Pérot interferometer is made up of two separated parallel reflecting surfaces; the interference happens due to the multiple superpositions of the reflected and transmitted beams of light in these parallel surfaces [36].

**Figure 5 biosensors-11-00058-f005:**
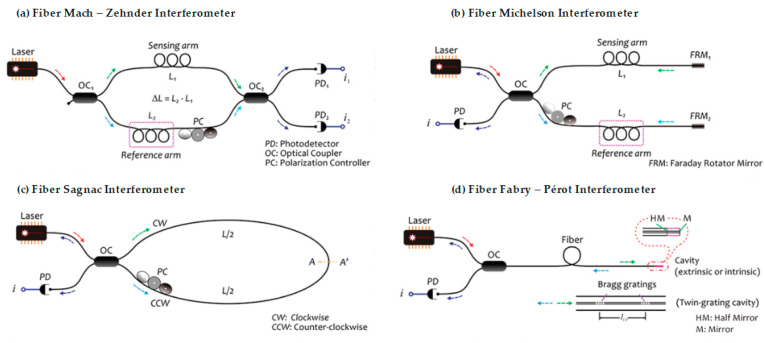
Configuration of different optical fiber interferometers [37]. (**a**) The figure exemplifies the basic operation of a fiber Mach–Zehnder interferometer (MZI) type, from the exit of the beam of the light source to its analysis through the photodetectors. (**b**) The figure shows a sensor using a fiber Michelson interferometer, which if analyzed is very similar in configuration to MZI, save for the fewer rotatory devices and mirrors. (**c**) The figure shows the configuration of the fiber Sagnac interferometer, where the opposite ways that the beam takes from light are observed with clarity. (**d**) The figure shows a fiber FPI type and the way in which the light beam travels from the source to its return to photodetector.

In Figure 6 are different configurations of as much an extrinsic as an intrinsic FPI sensors type, which is important since it shows us with greater detail and clarity the form in which the optical fiber can be characterized so that it works like a sensor. Here, it is possible to indicate that each interferometer has its different cavity configurations like: SMF with internal and external mirror, FBG cavity, SMF-Photonic Crystal Fiber(PCF)-SMF there are designed form measurement different signals or applications.

In Figure 7, showed some most common configurations on the fiber MZI sensors type, the changes in the microstructure of the fiber are by taper, joints with other fibers, and displacements of the cores between one or more fibers or with multicore fibers, where the fundamental issue is to make a difference of optical methods through the interference.

## 3. Health Field Sensing

In this section we describe and analyze according to their characteristics the research on OF sensors that seems to us the most important, which we highlight in each of the tables on each vital sign and where we compare the characteristics, operating ranges, sensitivity, and technology applied to the sensor. All this will help us to give a more accurate perspective of how most of them are good sensors, but they are still far from being able to operate as personal device or gadget in the home.

Cardiovascular diseases are a public health problem and the number one cause of death in the world [44], and for this very reason the constant monitoring of vital signs is important since many of the keys to this problem such as heart rate and respiratory rate, blood pressure, volume measurement, and oxygenation have specific applications and methods for monitoring. An example is the measurement of systolic pumping capacity, which is an important parameter for patients with congestive heart failure. Another example is when the heart stroke volume is multiplied by the heart rate, since it is possible to obtain the cardiac output, which is defined as the total volume of blood pumped per minute. Due to a reduced pumping capacity of the heart, patients with heart failure have a decreased cardiac output [45,46]. Like these examples, there is a wide variety of conditions that the human body presents due to cardiovascular problems, which, if detected in time and with constant monitoring, can be overcome or even eliminated.

The normal range of body temperature is between 36.5 °C and 37.3 °C. Temperature can be measured by tympanic route, orally, axillary, or rectally. Elevated temperature can be a sign of infection, arthritis, heat stroke, etc. Some useful temperature terms are hypothermia, which is a reduced body temperature of 35 °C or less. Pyrexia which is an elevated temperature, the three types of pyrexia are low (normal temperature up to 38 °C), moderate/high (38 to 40 °C) and hyperpyrexia (40 °C and above) [47]. 

Normal pulse or heart rate (HR) is between 60 and 100 beats per minute (BPM) in persons older than 10 years. In children and infants, the pulse is faster. Figure 8, shows the most common locations for easy acquisition of cardiac pulse and blood pressure. Several factors can affect pulse rate, such as fever, heart problems, infections, pyrexia, hypovolemia, hypovolemia, physical condition, anxiety, and medications. Some useful terms are tachycardia, a resting pulse greater than 100 BPM in adults, and bradycardia, a pulse less than 60 BPM [47].

For respiration, the normal respiratory rate is between 12 and 18 breaths per minute (bpm). Some of the factors that affect respiratory rate are anemia, pneumonia, asthma, chronic obstructive pulmonary disease (COPD), severe bleeding, stress/anxiety, medications, etc. Useful terminology includes bradypnea, or breathing slower than normal; tachypnea, or breathing faster than normal (greater than 20 bpm); dyspnea, or difficult breathing in conscious patients; orthopnea, dyspnea that occurs when the patient is lying down; and apnea, the temporary suspension of breathing [47].

For blood pressure, normal data for an adult human at rest is a blood pressure of 90/60 (systolic/diastolic) mmHg up to 120/80 mmHg; therefore, hypertension is blood pressure above 140/90mmHg. Figure 9 shows a scale of blood pressure values for adults, according to what is considered normal in systolic and diastolic pressure as well as the minimum and maximum limits (the ideal pressure represented in the green colored positions). The main causes of hypertension are obesity, chronic kidney disease, high alcohol consumption, smoking, and adrenal/thyroid disorders. Hypotension is low blood pressure (systolic less than 90 mmHg). Causes include pregnancy, dehydration, underactive thyroid, heart failure, blood loss, and anaphylaxis. [47].

### 3.1. Body Temperature

The body temperature of a healthy adult at rest is between 36.5°C and 37.3°C [47], with an average of 37 °C. In most of the articles that were analyzed, a clear tendency to use SMFs for the characterization of OF sensors to measure temperature was observed. Additionally, analyses of experiments within the near infrared spectrum (NIR), working mostly between 1500 nm and 1600 nm, facilitated the use of FBG to work with those periods and wavelengths. The sensitivity of results changes according to the other elements and techniques used for the creation of the respective sensor [51]. The research proposed a temperature sensor in biological tissues using an SMF, with the photo-thermal deflection of a laser that travels through an OF and its temperature gradient. Measurements have been taken in chicken organs (heart, liver, and gizzard) and where they have managed to obtain a heat pulse with an output power, all working at 1550 nm. Continuing with a project where researchers worked with an SMF at 1561 nm [14], they tested a sensor for human breathing detection that also used an FBG for temperature during the respiration monitoring process and measurements performed on the nose for obtaining sensitivity results of 11.4 pm/°C between 10 °C and 44.8 °C. A good research study was [52], in which they exposed a fiber sensor with possible applications in biomedicine, immunology, and biophysics, based on FBG working within the visible spectrum at 673nm with an SMF and using several types of lasers such as argon-ion, argon fluoride, and titanium–sapphire as a laser amplifier, finding a very wide temperature range between 242.45 °C and −211.55 °C. In another work [53], we found that researchers used the spectrum ranging from 1500 nm to 1600 nm; their sensor was an FBG to measure temperature during human mechanical ventilation, and the sensor was placed in an invasive mechanical ventilation tube coated with agar and polyamide acrylate: they obtained a total error of 3%, where the percentage represented the level of agar or agarose, obtaining sensitivity values of 114.7 pm/%, 0.12 nm/%, and 0.14 nm/%. Other work in which an SMF was first used to characterize the sensor obtained very large temperature ranges [54]: they presented a temperature sensor based on a mechanically induced fiber grating through a photoelastic effect with a heat-shrinkable tube, and its range of effectiveness within the spectrum was in the 1445 nm and 1570 nm, obtaining sensitivity results from de 3 nm/100 °C to 10 nm/100 °C, with a range between −20 °C and 50 °C. Continuing, [55], with a sensor where they showed its multiple possible applications but tested it with temperature, was based on an MZI with a pair of LPFG’s created with a CO_2_ laser, but measurements were performed in ethyl alcohol, with a wavelength of 1050 nm and 1239.4 nm, sensitivity of 11.7 pm/°C, and temperatures ranging from 30 °C to 110 °C. 

In Table 1, the most complete works are exhibited only on body temperature. The table’s columns identify the fiber optic type and operating wavelength in nanometers, where the sensor was tested, sensitivity in nanometers/Celsius degrees, characterization of the fiber optic nanostructure or utilization of doping agents, detection ranges on Celsius degrees, and the reference from where the information was taken. Additionally, the table shows that similarities can be clearly observed in the operating ranges of the first two articles [15,56], and the two [57,58] in the table using in all cases SMFs and the NIR spectra for their work. Based on the technology, two concatenated LPGs acting as an MZI for the design of a temperature refractometer [15] are presented to start with. In addition, a fiber sensor was found with possible applications for strain, temperature, and curvature. With characterization of the same with an LPFG through electrical discharges, a photonic OF is spliced with SMF, and the measurements are performed in the environment (Figure 10) [56]. In Figure 11, we can see a similar work, also used to join different types of fiber, the union of the fibers can be done through electric arcs or by means of some chemical compound. The entries continue with a humidity and temperature sensor characterized with only an LPFG created by an argon laser source [57]. Furthermore, another entry presents a very complex arrangement on an optical sensor based on the splicing of two pieces of SMF with a coreless fiber between them and where there are LPFGs to measure the temperature [58]. Next, a very recent study is presented on a sensor with an MMF to measure temperature through the surface plasmon resonance (SPR) technique, with integration for its utility with a smartphone; here, the device presented a flash along with a camera for signal processing through an application for the smartphone [59]—a great work with many possibilities. Finally we must point out a different sensor [14], as it is able to measure both body temperature and respiratory rate; in relation to the temperature yields, it has quite favorable results, although they varies greatly over time as it is affected by external factors.

### 3.2. Respiration or Breathing Rate

This is the vital sign that most researchers study with fiber optic sensors; temperature also has many research works but not as a vital sign [64]. Several things can be highlighted, such as the large number of creative works when performing experiments, the characterization of the OF, and that the vast majority use SMF; however, there are a wide variety of doping agents that are used to increase sensitivity or techniques to apply torque, force, or expose the same core of the fiber. Splicing and discharging are also found as the most used techniques to modify the nanostructure of the OF [65]. However, almost all the works use lasers and the NIR spectrum, thus decreasing the complexity or variables that could affect and intervene in the experiments. Besides, the calculations are more accurate with reference to reality [66]. Large and complex studies were observed that leave the door open for possible diversification and mass production of these sensors and use within the home market in the near future, since their characteristics and designs would allow them to compete against the electronic devices for monitoring vital signs that are currently available or would serve together to strengthen the quality of the same. [67].

Taking into consideration that a healthy resting adult human has a respiratory rate of 12 to 18 breaths per minute [47] (premature infants have the highest respiratory rates at 40–60 breaths per minute). Here we can observe an OF sensor interferometer that measures nasal curvature during the respiratory process, through inhalation and exhalation, with an overall performance accuracy of 5.62% (with variations of ±2.26% in inhalation and ±3.19% in exhalation) and a sensitivity of 94.4% for inhalation and 93.93% for exhalation. [68]. They proposed a noninvasive OF sensor to monitor respiration, blood pressure, and movement in the human body using MMF and SMF. Another [69] presented a model OF sensor [70] with an interferometer for detecting heart pulse and respiration by putting it on a mattress, where the measurements ranged from 1561.07 nm to 1561.467 nm. Continuing, a work was shown for the monitoring of respiration through a FBG sensor [71] with a plastic optical fiber (POF) implanted in a flexible textile that measures abdominal curvature. The properties of polymer composite fibers make them unique and very good for their ease of handling, as they can withstand rough treatment compared to traditional fibers; they use a polymethylmethacrylate in the core of the plastic fiber and a fluoropolymer in the coating with wavelengths ranging from 650 nm to 655.9 nm, with a maximum error of 1 % with a very short fiber of 6.6 cm in length. Here, we continue analyzing [72], an FBG sensor to measure humidity, temperature, and respiration during the process of invasive mechanical ventilation through endotracheal intubation, obtaining BR readings from 11.5 breaths per minute (bpm) to 24 bpm. It can be observed (Figure 12) that its OF is given a layer of hygroscopic material, agar, and an acrylate cover for protection since the characterized fiber goes inside a hypodermic needle, achieving a confidence level during monitoring of 95%. Another project that draws attention was [73], where they presented an OF sensor generated with microcurves forming a mesh for the measurement of the perioperative pulse and respiratory rate in infants [74] from 01 to 12 months of age. Monitoring of the infant was performed through a mat where he remained lying down to obtain readings of 23.5 breaths per minute (bpm) to 23.5 bpm all with an MMF working in the 1310 nm of the NIR spectrum.

In Table 2, the most complete articles with respect to our parameters are highlighted but only those where the OFs used did not receive any modification in their nanostructure by any dopant or coupling agent. Heading the table [75] we find the proposal of a series of parallel optical sensors to measure respiration by means of thoracic and abdominal curvature during the respiratory process, using gratings with periods of 532 nm. The following article is a continuation of the same Allsop study from 2004, where they made measurements in the human rib cage and abdomen. Another work that presented an LPFG is one that monitored curvature during chest movements on breathing; it also measured the amount of air during stimulation and exhalation to make the results more accurate measurements of the chest and abdomen [76]. Additionally seen was a comparison of three sensors to measure continuous breathing during anesthesia application in a magnetic resonance imaging (MRI)) [77]; this allowed continuous sampling of the motion caused by breathing in the chest and abdomen of the human body [78]. There are even sensors with FBG to measure the volume of air during the breathing process by curvature of the chest [79], and there are also those but with fibers attached to tissues [80] that monitor almost the same movements during magnetic resonance imaging (MRI) examinations [81]. As a complex with 12 FBGs for monitoring respiratory parameters and heart rate (that is, the volumes and movements involved in the chest and abdomen), the minimal movement associated with human breathing is monitored throughout the system [82]. Some have been built to measure a particular element or substance but have been successful in detecting human respiration through testing in other areas. Here they use a temperature FBG in the breathing monitoring process. [83]. Figure 13, shows a representation of smart textiles, where the aim is to weave OF into clothing and present design alternatives so that they can be in contact with the skin and do their monitoring work. This is one of the great challenges of smart textiles with OF sensors.

Continuing with a study where [86] showed an optical balistocardiogram technique to measure cardiac and respiratory activity noninvasively, they used an optical FBG device with a germanium-doped SMF placed on the thorax, obtaining sensitivity of 1.20 pm (με)^−1^, corresponding to 17 Min^−1^. They presented an SMF sensor based on BFG, where several methods were proposed for obtaining vital signs [87], as well as modifications for the characterization of the same fiber with doping agents such as germanium, indium arsenide, and gallium; measurements were performed at the waist, neck, elbow, and ankle of the human body as they sought to determine where it was best to place their sensor. They obtained detection ranges of 1550 ± 0.5 and 1525–1570 nm ± 0.1, or the wavelength at which it operated. In another work, we can observe in Figure 14 how through the characterization of an SMF they created a twin core that operated from 1551.85 nm to 1552.15 nm, where they used a dopant agent of pulverized gold [88]. The result was a noninvasive sensor to measure respiration and HR with readings in the human chest, obtaining a sensitivity of 18 nm/m^−1^.

In Table 3, we present data from work done with respiration but which involved a modification in the nanostructure of the OF by using chemical doping agents, which is why we differentiate them from the studies in Table 2—apart from the fact that it can be observed they are the most investigated fiber optic sensors, with the highest number of studies that are talking about vital signs.

An LPFG sensor with three SMF fibers that analyzes the curvature and heat generated during breath monitoring was observed, even though the measurements were performed on a resuscitation training dummy—on the chest [79]. Using FBG, there was a sensor encapsulated in a compound with excellent thermal and elastic properties to monitor respiration and measure data from the sensor placed on the human chest [89]. Continuing with a study of a fiber optic sensor with bends for breathing monitoring and a sensor on the back and chest [90], so far it can be seen that almost all studies on breathing with OF were achieved through the measurements they obtained on the chest and the chest during the same process of human breathing—something logical and relatively simple since it is just a count of the breaths that occur in a given time. Another work is that of an FBG sensor to measure respiration and cardiac pulse during MRI examinations; as the sensor is immune to electromagnetic interference, the measurements were performed on the body through a mattress for a patient in supine position [91]. Likewise, a really striking and creative sensor is presented as is a smart textile, composed of 12 FBG sensors to measure all trunk movements, including breathing; being a whole garment, its measurements were obtained in chest, rib cage, and human abdomen [83] (Figure 15, shows various representations of smart textiles with OF sensors). Finally there is a noninvasive device (Figure 16) for respiratory monitoring through a fiber with two FBG, which measures nasal airflow [92].

### 3.3. Pulse or Heart Rate

With the following vital sign, it is clear to us from the studies presented and the literature that pulse or heart rate is almost always accompanied by some other vital sign, either body temperature or respiratory rate, since the similarity for obtaining it through pressure or temperature (more common) is the same and only the parameters for its interpretation or mathematical calculations based on the heartbeat change. However, the usefulness of the sensor or characterization changes completely when we analyze the techniques in depth; although they are similar, they all have small details that identify them and give a sensor its unique character. Here we also begin to glimpse the problem of replication and mass production that would be desirable due to the size and characteristics of the sensors. A healthy adult at rest has on average a HR between 60 and 100 beats per minute (BPM) [47]. We can also say that a good way to know the physical health of an individual is through the monitoring and measurement of the heart rate, since this way it is possible to know the intensity and physical work of an individual, strength, expenditure, and calorie intake [92], among other things—in addition to how essential it is for the diagnosis of diseases, constant monitoring and treatment of people efficiently [97]. FBG-based sensors have been studied in processes. Since it is possible to establish the pressure needed during the perforation of the cardiac wall, in a study of cardiac arrhythmias, in vivo tests were performed in the hearts of two sheep. Working with an SMF in the range of 1561 nm to 1564 nm, an invasive procedure was used where the sensor was protected with a steel cylinder and urethane with epoxy glue [98]. Another way to find the cardiac pulse is through an optical sensor based on the reflectivity of light, where its change can be measured through the movement of the chest. This was done with a POF (plastic optical fiber) based on the principles of light intensity change every time the heart emits some vibration. With these studies it was possible to detect the peaks of a harmonic signal at a frequency of 15 Hz, 7.5 Hz, 10.5 Hz, and 22.5 Hz in a healthy adult with a mean value of 78 BPM; these results were so promising that the researchers made a working prototype [99]. 

Starting with the analysis of the most complete work, the best elements for the equipment are those presented in Table 4. First, a physical sensor that measured the heart pulse through a pillow placed on the back of the head will be analyzed; to improve the sensitivity, microbends were made in the fiber. It was found that the accuracy of the sensor decreased as the weight of the head increased. To obtain the results, the same sensor was compared with a commercial sensor [100]. The following is a paper in which they present an FBG sensor for the detection of HR through vibrations: the sensor was capable of being used during MRI procedures due to its low signal-to-noise ratio and being immune to electromagnetic interference [101]. Another study is one in which they showed the development of a FBG sensor for monitoring vital signs; this was performed superficially in different parts of the body, where different results were obtained compared to other sensors using the techniques of Bland Altman. Pulse and blood pressure were finally obtained by performing calculations and measurements in the temple, finger, ankle, and dorsum of the foot. This turned out to be a very complete work where different parts of the body were taken into account and where it was possible to measure the HR correctly [102]. We can also observe another fiber optic sensor with FBG for HR monitoring, where the use of an elastomeric material with measurements on the chest was tested. This type of material is very useful in the field of health and medicine because its properties are very friendly to human skin [12]. Another team showed a fiber sensor to measure HR, where patients were placed in a massage chair to monitor the data that were delivered by a pad on their resting heads—an effective method and also very useful because it helped to relax and lower the HR [103]. A very complete work is that of an OF sensor based on FBG where several methods for obtaining vital signs were proposed, as well as modifications for the characterization of the fiber with measurements in the waist, neck, elbow, and ankle of the human body; here, a measurement of temperature obtained during the exhalation process was performed [87]. This work is a clear example of the possibility and feasibility of measuring another vital sign during the process of obtaining the one that really interests us—it could well be by the hand of some other electronic or auxiliary device. Finally, an HR sensor made with a POF with three different configurations of the same fiber—which were straight, sinusoidal, and spiral-shaped, and were tested on the human hand, chest, and neck under fast, normal, and slow HR conditions—highlighted the photo elastic properties of the fiber [104]. In Figure 17 we can see an example of the work of Bonefacino, J. [60], where through an FBG sensor they managed to obtain measurements of the human pulse and respiration, all through an ultra-fast etching process of the strips. They showed sensitivity data of 150 bpm in HR and 8 BPM in BR.

Figure 18 also shows an example of the work done by Chuanglu, C. [107], where you can see that through the fitting of a signal obtained by a FBG sensor and smoothing this same signal, you can get the desired results. Here is shown the behavior of the same blood pressure (systole/diastolic) with a device that can also get more data, such as respiration and pulse.

### 3.4. Arterial or Blood Pressure

Finally, we have the vital sign of blood pressure, where an adult human at rest has a blood pressure of 90/60 mmHg to 120/80 mmHg [47]. Blood pressure turns out to be the most complicated vital sign to obtain through OF sensors since in a traditional way it is very easy to apply pressure on the arm through a sphygmomanometer or tensiometer and thus stop the flow, which will cause an increase in pressure that can be measured. This same principle applied to fiber optic sensors is complicated since by increasing the sensitivity and characterization of the OF, these data can be confused with BR or HR but, through calculations or an algorithm is easier to obtain, resulting in the best way to obtain these data. Blood pressure also serves as a way to know or estimate multiple causes of cardiovascular conditions or problems. To start, we have research on the feasibility of using a system based on OF for invasive measurement of blood pressure [108] in which they present gain and offset errors of <3.4% and <0.25%, respectively, delay < 1 msec, and maximum rate of change > 30,000 mm Hg/sec, and where the effectiveness of using an OF to withstand electromagnetic interference was obtained. Another example is the study of a prototype sensor to measure Doppler blood flow velocity through a SMF [109] working in the NIR at 790 nm and 25 °C, where the Doppler shift of the radiation of a laser was recorded with an OF, thus beginning to visualize the intervention of possible tools for medical aid or assistance with the characterization of optical sensors.

Starting with the research work presented in Table 5, we have firstly an OF sensor of FPI for medical use (blood pressure) that works with white light; experiments and measurements were performed in a goat in which the internal pressure changes of the heart and the aortic vein were monitored [110]. Next, a work where they present another OF sensor to measure blood pressure in vivo and invasively, using as a sample a pig in which a catheter was introduced into its coronary artery from which blood pressure measurements were obtained in the aortic arch, right coronary artery [111]. We continue with an FBG sensor that measures waves in the deformations of the body surface that turned out to be pulse waves that were generated with the contractions and expansions of the arteries, achieving reliable values on the systolic pressure, where measurements were obtained from the neck and ankles of the human body [112]. Then, we have an OF sensor that looks like an advance from a previous sensor [111], where a complete design and packaging of the sensor was created for in vivo use in a swine coronary artery to measure the pressure of the arch of the aorta and the right coronary artery; it was demonstrated that the method of encapsulating the sensor provided effective protection and flexibility in the handling of the OF sensor in vivo for blood pressure measurements. Basically, it was a feasibility study of the operation of the sensor and how they provided protection and stability to work outside the laboratory—undoubtedly an excellent work as it provides evidence for the future possibility of further mass producing or developing OF sensors for the market [113]. We continue with the development of a FBG sensor for monitoring vital signs; this was performed superficially in different parts of the body, where different results were obtained compared with other sensors using the Bland–Altman technique. Pulse and blood pressure were finally obtained by performing calculations; the measurements were performed at different parts of the human body: temple, finger, ankle, and dorsum of the foot [102]. We also observed a portable sensor that used an FBG to measure tensions with high accuracy as the pulse wave signal; that is, together with mathematical calculations as the blood pressure values were obtained, the measurements were performed on the arm. This study validated that when the measurements are at different heights, the results change until the body posture changes. The measurement turned out to be good as long as the height was not changed—at least it was the parameter that affected the blood pressure sensor they had [114]. An FBG sensor in a smart textile made of silk fabric was presented for constant monitoring of vital signs, which can be calculated by analyzing the generated pulse wave, with special emphasis on blood pressure [115]. Another very good work (Figure 19) is from a study in which they showed an FBG sensor to measure HR and blood pressure through an aluminum diaphragm [105] that was placed on the wrist of the arm, which served as an acoustic amplifier when emitting such pulses and could be detected through the characterized fiber. This is a very good idea implemented for obtaining data through the generated sound waves and their analysis to generate results on the activity of pressure, pulse, and blood viscosity. Another group presented an FBG sensor with a POF [116]; in this case they proposed the use of plastic fiber since the common fiber is very delicate and could fracture during the operation of the sensor; therefore, with the POF that curl is omitted, the sensor gets to detect a pulse wave for blood pressure up to 8 times higher than with a silica OF. The results show that the blood pressure had very little error but the acceleration correlation of the plethysmograph was not the desired one. Finally, a sensor based on FBG is presented where several methods for obtaining vital signs were proposed, as well as modifications for fiber characterization. Measurement on the waist, neck, elbow, and ankle of the human body, the temperature measured was the one emanating from the exhalation process and the blood pressure was measured through the variations in the diameter of the artery caused by the same pressure. It was also concluded that the sensor could not effectively measure the pulse and pressure, one or the other should be separately eligible [87].

### 3.5. Multiparametric

These represent investigative works that are outstanding due to their complexity and their ability to monitor multiple vital signs or some complementary characteristic that serves to obtain the same signs [117]. Although most base their technique on obtaining pressure or temperature along with mathematical calculations to obtain the other vital signs, they do so by comparing their results with sensors that are already present in the market and trading through Bland–Altman techniques. 

In Table 6 we start by reviewing a work on an interferometer-type fiber sensor where different configurations of the same for monitoring psychophysical activity were compared to sample HR, motion, and blood pressure, performing measurements on the human body [118]. That study is followed by a proposed noninvasive OF sensor to monitor respiration, blood pressure, and movement with drugs on the human body through a chair with a backrest and a pillow where it carried the sensor [119]. Continuing with another FBG-based sensor to measure respiration, pulse, and movement, but also to obtain data on temperature variations, measurements were taken on the human neck of a sample subject [61]. Next, a work was analyzed where a sensor that handled an FBG with microcurves is presented showing its practicality in measuring heart rate and respiratory rate. This sensor with multiple uses and capabilities through smart objects was tested using mattresses, pillows, chairs, and swings where a sample subject was placed [84]. The review continues with a proposal on a ballistocardiogram technique using optics to measure cardiac and respiratory activity noninvasively—using an optical FBG device with measurements on the human chest [86]. Next is a design with its verified functionality of a noninvasive sensor to measure and monitor pulse and respiratory rate through an FBG measuring rib cage movements and also performing measurements on the human chest [93]. Work continues on a noninvasive multichannel OF sensor for HR, BR, and temperature monitoring through FBG characterization, encapsulated in a polydimethylsiloxane polymer where measurements were performed on the human rib cage [62]. Then an FBG sensor integrating three types of sensing for pulse, HR, and body temperature is analyzed. The study consisted measurements on the human body using a fairly complete vital signs sensor based of OF and was developed by a team of researchers [69]. Our review continues with a sensor for the detection of human respiration that used an FBG for temperature during the process of monitoring respiration, making measurements in the nose [14]. Another FBG sensor for simultaneous measurement of respiration used pulse wave and heart sound with measurements at the tricuspid region of the chest and the carotid artery of the neck [7]. Other techniques created FBGs in an SMF and used them in the biomedical field as HR, BR, and temperature sensors, performing measurements in the human chest. Additionally, techniques are shown to increase the sensitivity of the fibers using different doping agents for the fiber and also exposure to ultraviolet rays [60]. There is also a proposed model on a fiber optic sensor with interferometer for the detection of heart pulse and respiration through its tuning on a mat to measure different parts of the body [69]. The review continues with consideration of a sensor for monitoring heart rate and respiratory rate through a design of a ballistocardiogram composed of FBG—all to be applied during the MRI process where it was shown that the sensor was not affected by any disturbance caused by the resonance analysis, this through measurements on the human chest [63]. Finally, we have a work that showed a sensor with a FBG to measure the heart pulse and blood pressure through an aluminum diaphragm that was placed on the wrist of the body, which served as an acoustic amplifier at the time of emitting such pulses and was able to detect through the characterized fiber [105].

Table 6 only shows those works on fiber optic sensors that were capable of obtaining measurements of two or more vital signs; however, for ease of comparison, the data of most interest to us were included in the corresponding tables.

## 4. Discussion

The preceding provides us with an important starting point and a research base to continue testing more features in sensors that have a competitive potential in the market or can be used to generate knowledge. There is a growing demand for medical devices for constantly monitoring vital signs, health, and stability as a means to prevent deterioration of the patient’s condition or the illness, or simply to warn of any elevation that may indicate a more serious disease. This leads to the need for medical sensor application technology in real-time, at low cost, and that is commercially viable, that compete with devices commercial electronics.

A large number of the sensors in the literature reviewed work with FBG-characterized SMFs operating at a wavelength within the NIR (1550 nm), where they are more sensitive to the change in the refractive index of the OF and are for monitoring respiratory rate, as it is the most commonly used, inexpensive, and therefore most accessible fiber type. Apart from the fact that the characterization with FBG at that wavelength is a widely tested that has given multiple results using lasers [120], the complexity is to adjust the sensor as a filter fort to measurement of some vital sign [72,114,115]. All the sensors analyzed, are performed for measurements of physical parameters in the medical field (temperature, position, force, torque, stretch) but is necessary its calibration and categorization, as they are able to perform different measurements depending of the configuration and design. Therefore, they are applied for obtained manage and to obtain a response of vital signs [87,121].

Unlike the systems of traditional detection and monitoring of vital signs, optical fiber sensors have very attractive characteristics, such as excellent sensitivity, great rank of operation, and a high trustworthiness. The advantage of optical fibers use is that they are elaborated with dielectric materials that are chemically inert (that is, compatible and immune to the electromagnetic interference) and many of them can be covered by designed materials to support great temperatures. These properties make them excellent so that in their characterization as sensors, they can be adapted in hostile and aggressive surroundings, such as the interior of some block of construction, materials, or structures, or in a system of generation and transmission of electricity, where conventional sensors can have an unstable operation and at the same time have increases in the possibility of faults [37]. According to the data that appear in the different tables, one can deduce that the measurement or monitoring of vital signs varies depending on the atmosphere where the registries are made. With almost all reported results, were tested in a laboratory. The tests would be much better in real environments since many sensors are very sensitive to pressure, temperature and vibration changes. It would be ideal if the designs that were investigated could be developed as prototypes. In the review of the reported research works, we found the lack of classification of those sensors that reached an operational sensitivity for applications in health care. Because, each research team established its criteria for the characterization of the sensor, such as: units of measure and limit of operation. Since there is no standard characterization criterion, it is difficult to establish with certainty the feasibility of operation of all sensors of each research team. Knowing only the use and operation of the sensor that was considered under certain very specific conditions.

The results obtained in some works show an enormous variation solely with changing the area of positioning of the sensor. For example, in the case of Chino, S. et al. [102], data varied depending on where the samples were obtained from the four parts of the body (temple, finger, ankle or dorsum pedis) for the obtaining cardiac pulse. This is evidence that, if they wanted to obtain good results for locating some of the vital signs, it is necessary to study the medical literature on the different vital signs, such as physiology, which firstly allows to establish parameters for the correct taking of samples and tests.

The sensors in Table 6 are the most robust and complete proposals for measuring vital signs; their sensitivity and reliability make them great candidates for operation in in vivo situations; however, for us the best are still those that do not require human intervention, but those that are placed on some kind of guide, mattress, or pad [69,74,84,91,103,119], since the error factor due to direct human intervention and the practicality of handling the sensors themselves is eliminated. It can be clearly observed in this table that cardiac pulse and respiration are almost always obtained together. Another important characteristic is to see that only in one of them the blood pressure is obtained together with the cardiac pulse. Finally, another characteristic is that although some are large sensor arrays, not having more sensors means that their capacities grow; the number of elements or sensors involved in each research study does not determine their accuracy, reliability, or greater capacities.

The complexity of the sensors lies not only in the characterization of each one of them, but in that there are some [82,92] in which the monitoring is performed through several OFs, different types of OFs, or with several sensors in the same array. These represent complex models if we take into account their nanostructure and fragility to be handled without suffering any damage, since portability is one of their major disadvantages, without considering their light source, since in most of the researcy they turn out to be lasers. 

There are also studies that try to analyze [12,62,72,89,105,113,115] the means of transport or protection, since the tests have brought these means to a more practical position, which is desirable if you want to develop and install such sensors in the market, since the tests ground them to a real situation—which is convenient if what is sought is the development and implementation of this type of sensors. It should be noted that OF sensors already exist commercially, but their use is still limited.

We believe that the best works were those in which they could remain as a prototype proposal [99] and were implemented through comparisons by Bland–Altman techniques [102] against others for commercial use [100] or they were presented and put into operation suddenly.

The use of fiber optic sensors in conjunction with new technologies provides a solid structure for the design of new sensors. The use of characterized FO is of vital importance, since almost all research papers document improved sensitivity, managing to overcome Bland–Altman tests against established devices or gold standards to determine sensitivity and operation during laboratory tests. This not only for vital signs [63,91,105,115] but also for a wide variety of optical sensors for use in the medical field, such as bacteria sensors [122], glucose monitors [123], cardiovascular monitors [107] (although this research work fails to present conclusive results due to the small number of samples, but they prove to have better sensitivity values), diabetic foot treatment [124], and also in the field of biochemistry [125] as diagnostic sensors, environmental monitoring, and organic and chemical compounds.

These sensors demonstrate their great effectiveness, presenting innovative and difficult to overcome designs since the characterization of the optical fibers can deliver extreme sensitivity and accurate data. In terms of cost, they have not been able to be introduced as sensors for domestic use since they cannot compete in cost relative to the established electronic vital sign sensors. The trend observed is the integration of these sensors with different devices, such as programmable boards for hardware development (Arduino and Raspberry Pi) that combine the best of each technology, such as miniaturization, free access, high sensitivity, and immunity to electromagnetic interference, among others—which combined have proven to be sensors with exceptional capabilities.

In addition, in some cases they show similar levels of sensitivity to fibers used in some high cost articles; that is to say, an alternative of lower cost is appearing, but even so it is not sufficiently attractive when competing with the traditional vital signs sensing devices [126,127,128] that often seem to be more precise and reliable. On the other hand, some otherwise very good work has achieved unexpected results [90], since the percentage of error or variation was very high, due to factors such as temperature differences or pressure. Nevertheless, being good studies, they continue to indicate methodologies to follow or to correct for future projects. 

With respect to the sensors for human temperature, from all the work presented in Table 1, we can conclude that the best in terms of operating ranges [6,60,61,62] are those that only measure the ranges proposed for human temperature (36.5 °C and 37.3 °C), since many are able to detect very high temperatures and are really surplus. In terms of sensitivity, all of them have very small ranges (surface/temperature), from which one could well infer their accuracy. Finally, none were placed in the most commonly used parts of the human body to obtain the temperature, such as the armpit, forehead, ear, or rectum.; we infer that the lack of testing in these sites is because they require presenting a sensor, rather than proposing manipulation or live operation.

With respect to the sensors for monitoring respiration presented in Table 2 and Table 3, we conclude the following: the error range present could only be acceptable if there were a standard metric to compare all the sensors, since in many cases it is only a simple counting of breaths, while in others the signal is generated by each movement detected by the sensor and analyzed. However, the best are those that have very little variation or error range [58,60,63,65,66,71]. Here we can conclude that, in effect, it is something very difficult to calculate due to the physical activity of the sample subject and, as in the temperature, only sensors and their operation are presented, not an arrangement that allows to take them into real situations for practice or operation—places working against the environment and its variables.

With respect to the sensors for cardiac pulse monitoring presented in Table 4, we conclude the following: the best ones are those whose data conform to the standards (60 BPM to 100 BPM) and whose degree of error or variation is the least [8,93,129]. Taking into account these data, we can assume that some of them do not present maximum and minimum values of operation, or their respective values of error or standard deviation. 

For the sensors for blood pressure monitoring presented in Table 5, we conclude: the best results [88,89,90,91] are for those that show greater test data due to the different zones of the body where the sensor was placed. This is because a change in blood pressure can be seen in these areas, where it can be said that the best area is the wrist, due to the ease of placement of the sensor and the results shown. The only desirable thing would be to test and prove the feasibility of these sensors in extreme conditions of physical activity (walking or some exercise) and to observe any variation.

## 5. Conclusions

Fiber optic sensors have great advantages, such as being immune to radiation and electromagnetic interference, chemical corrosion. Their lightness and small size are attributes that make them perfect for monitoring health or for use in the medical field as a great tool capable of providing information in real time—even more so with the monitoring of vital signs, where these sensors can be used as a simple monitor or even life support for the diagnosis or prevention of disease.

The projection of Fiber Optic Sensors [130], has grown very slowly into the markets directly competing with conventional sensor technology from 1980 to 2000, due largely to the high cost and a limited number of sensors. Considering the evolution of fiber optic sensors and a projection for the year 2020, it is expected that the cost of components for their manufacture will be more economical and attractive. This would result in a greater opening and distribution of the market, more economical and affordable sensors for companies, and better quality components.

The trend of our time is the preparation of more accurate and sensitive, faster, and highly specific fiber sensors, not only in the field of medicine and health. On the other hand, these sensors have a tendency to be more complex and therefore financially demanding, making them more expensive to mass produce. The number of publications on sensors for medical and health care has increased in recent years and without a doubt this will grow enormously in 2021 due to the problems that humanity faces with COVID-19. We are at a decisive moment where the constant monitoring of our vital signs must be taken into account and we must be attentive to any changes; in fact, the, respiratory rate monitors and oximeters have seen their commercial demand increase enormously since the vast majority of people who have the possibility of acquiring any of them have no choice but to monitor their vital signs in isolation due to the possibility of contagion of the disease. It has become common to measure temperature in all meeting establishments or places of human recreation. All this favors that these sensors can ultimately see a commercial outlet and not just remain as laboratory projects. Although there are some that as prototypes seem to be very good, one can perhaps find a way to make them more resilient or to improve their portability or to use visible light—since one of the technolgy’s main problems is its fragility and light source. 

In this article, a review of fiber optic sensors for monitoring vital signs was conducted, where characteristics such as the technologies used for manufacture, compounds or elements that modify the nanostructure of the optical fiber, operating ranges, and sensitivity are presented and discussed. We analyzed which of them are the best proposals for their possible live operation, since it is one of the concerns to be uncovered with this work. The main feature of all of them is the monitoring of vital signs in real time. Some of them even measure oxygen saturation and stress. Some very ingenious or complex designs are illustrated, both as their insertion in textiles and arrangements with various types of fiber—each of them with different modifications in their cores. In conclusion, fiber optic sensors for monitoring vital signs allow the technology enclosed in them to design prototypes for use in various healthcare facilities. These sensors greatly improve the comfort, ease of operation, and accuracy of health monitoring devices.

## Figures and Tables

**Figure 1 biosensors-11-00058-f001:**
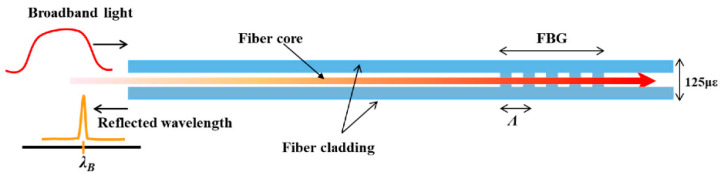
Graphical mode of operation of a fiber Bragg grating (FBG) sensor [19]. In this type of sensor, the grating is inserted into the core of the optical fiber (OF), as shown in a signal where the wavelength filtering is done by such a sensor.

**Figure 3 biosensors-11-00058-f003:**
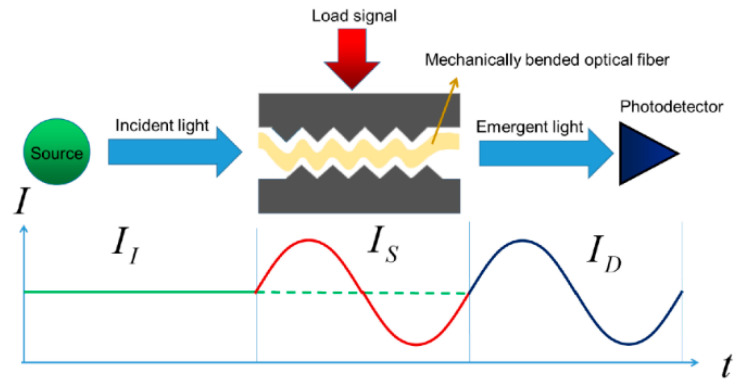
Schematic of a microbend fiber sensor bay mechanical induction [28]. The figure shows the mechanism of how the fiber undergoes microbending to create a sensor and microbending losses caused by disturbances in the optical fiber.

**Figure 6 biosensors-11-00058-f006:**
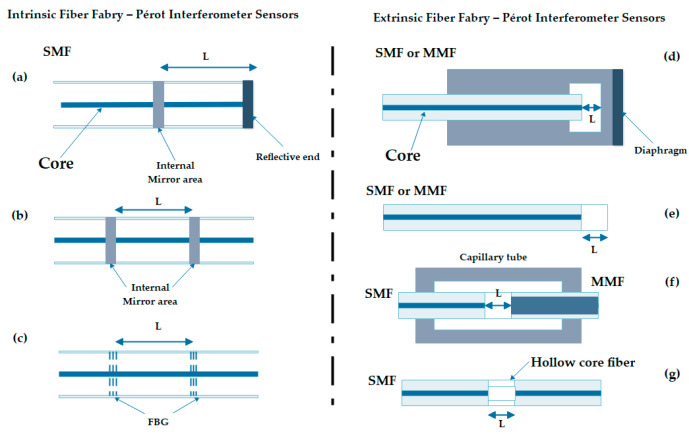
Configurations of FPI sensors [38]. The (adapted) figure shows the intrinsic and extrinsic configurations of FPI fiber sensors, where (**a**) represents a cavity formed by an internal mirror at the end of a fiber; (**b**) a cavity formed by two internal mirrors; and (**c**) a cavity formed by two FBGs. For each case, L represents the length of the optical cavity. Configuration (**d**) represents a cavity formed by a diaphragm at the end of a fiber; (**e**) a cavity formed by the surfaces of a cover in the end of the fiber; (**f**) a cavity formed in the end of a single-mode optical fiber (SMF) and an aligned multi-mode fiber (MMF) through a capillary; (**g**) a cavity formed by a SMF that the end is joined with a hollow core fiber. Additionally, the equal one in each case L represents the length of the optical cavity, which is an air bubble in the configurations (**d**), (**f**), and (**g**).

**Figure 7 biosensors-11-00058-f007:**
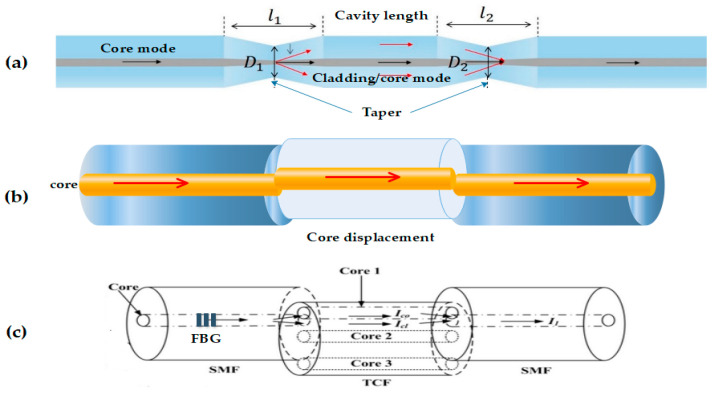
Different configurations from MZI fiber sensor (adapted) [39,40]. Here are different fiber MZI where we point out: (**a**) shows an interferometer created through a taper in the optical fiber, the same that is obtained by applying force in opposite directions when an unloading by electrical arc is made to modify the nano structure of optical fiber [41]; (**b**) shows joints of optical fiber but with a space displacement in the cores, which originates interference [42]; (**c**) shows joints of several types of optical fiber, where in the first, a fiber Bragg grating type created on a single mode fiber that is joined with an optical fiber with three cores that also hold a concealed knife with another single mode fiber segment [43]. The above are different techniques that are used to generate the sensors and, like the configurations of Fabry–Pérot, these are made to obtain greater sensitivity, specificity, stability, or exactitude in the sensor that one wants to create.

**Figure 8 biosensors-11-00058-f008:**
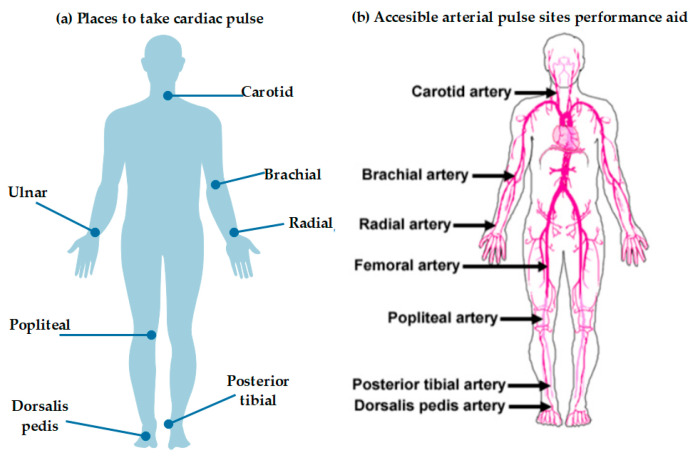
(adapted) [48] Human body areas for heart rate and blood pressure measurement. In (**a**) are the main areas where the cardiac pulse can be taken in the human body, since it is there where the main arteries are located and therefore the measurements are easier and more accurate; (**b**) shows the most accessible places for sampling blood pressure; it is observed that almost the same arteries are used for blood pressure and cardiac pulse, so that some sensors are able to perform both samples through the same signal [49].

**Figure 9 biosensors-11-00058-f009:**
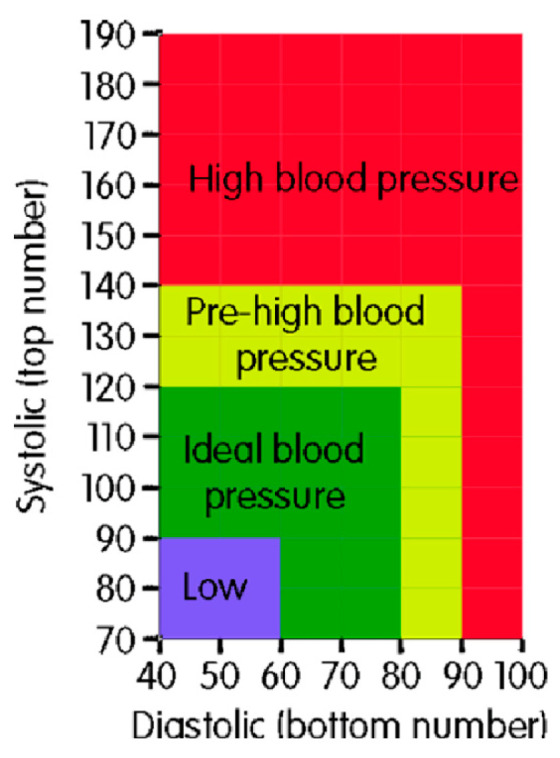
Main ranges for blood pressure [50]. The figure shows the ranges of blood pressure in adults, ranging from low, ideal, acceptable high, and severe high, both systolic and diastolic pressure.

**Figure 10 biosensors-11-00058-f010:**
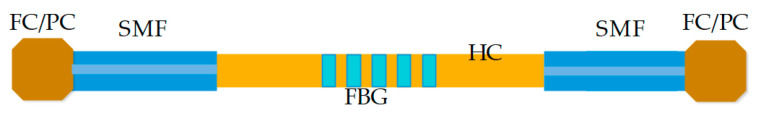
(Adapted) Schematic design of the sensor proposed by Agostino Iadicicco. A view of the hollow core fiber is displayed with an LPFG and in turn spliced with conventional SMF and their respective fiber optic / physical contact (FC/PC) connectors [56]. The use of multiple splices together with different fiber types is common to obtain higher sensitivity or sensors capable of high-precision filtering; however, the right materials and tools are needed to achieve so much precision in the OF nanostructure.

**Figure 11 biosensors-11-00058-f011:**
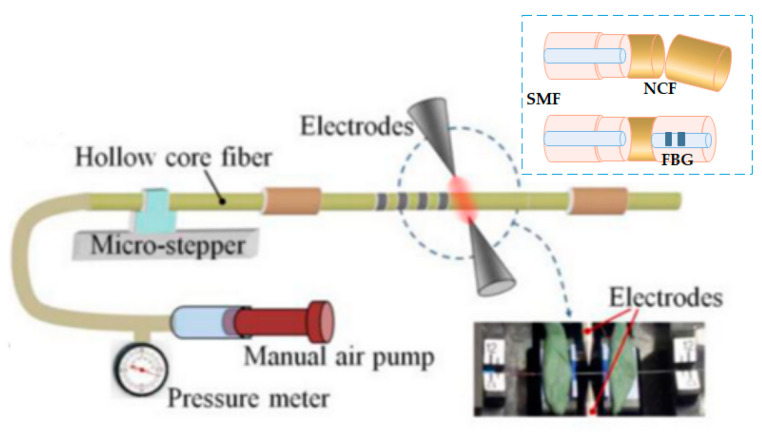
**(Adaptation)** Example of sensor non-core fiber – long period grating (SNS-LPG) experimental arrangement proposed by Agostino Iadicicco. This is the experimental arrangement they propose for splicing two SMF parts with a non-core fiber (NCF) in the middle and there create a LPBG, resulting in the SNS-LPG arrangement [56]. This is another work that hints at the accuracy of the equipment to characterize OF nanostructure; no doubt these are innovative ideas, but outside of a laboratory for our purposes they are not feasible—they are rather sensors for research.

**Figure 12 biosensors-11-00058-f012:**
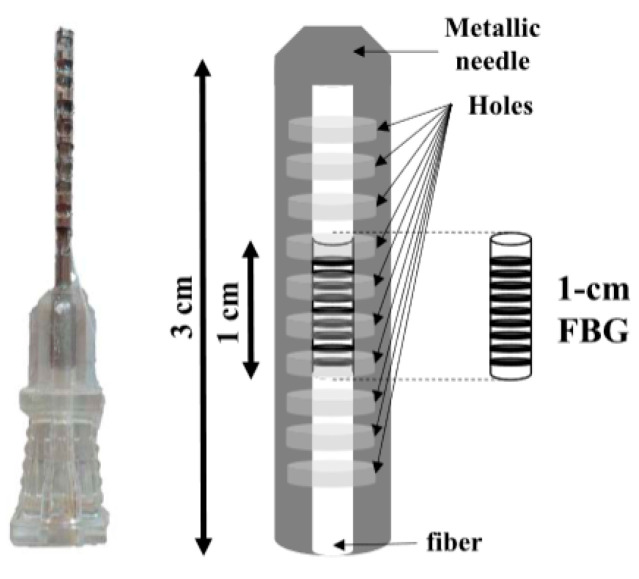
C. Massaroni guide and sensor protection scheme. This is an example of the hypodermic needle. In the ventilation process of mechanical invasion, the hypodermic needle is used to measure respiration, temperature, and humidity. In addition, with a really simple and innovative design, the hypodermic needle is used to protect the OF [72].

**Figure 13 biosensors-11-00058-f013:**
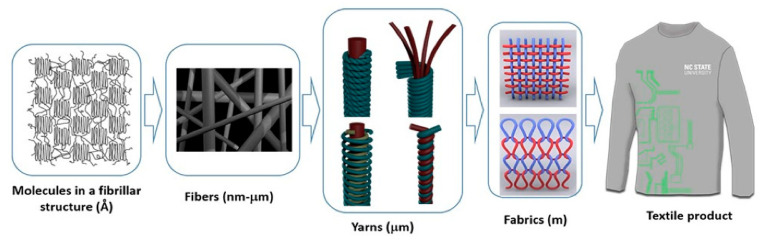
OF textile material representation by Kony Chatterjee. Here you can see the great work of twisting the optical fiber and weaving it to a textile, all for transport and protection, as it provides an ingenious way to move it and place it in a textile garment to be in constant contact with the sample patient. This is a clear example of how you could make different textile garments depending on what you want to monitor and the part of the human body from which you want to sample data [85].

**Figure 14 biosensors-11-00058-f014:**
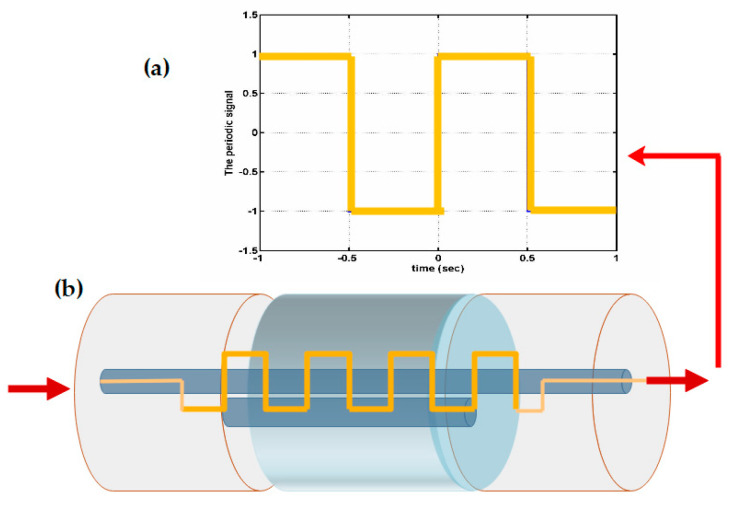
**(Adapted)** Design of the twin core fiber and its operation. In (**a**) you can also observe the operating wavelength of the SMF once it is characterized as a twin core, where it is also doped with sprayed gold to increase sensitivity and in (**b**) the schematic design and representation of the characterized fiber—a great job with a lot of creativity to create single mode fiber—two core fiber—single mode fiber (SMF-TCF-SMF) array [88].

**Figure 15 biosensors-11-00058-f015:**
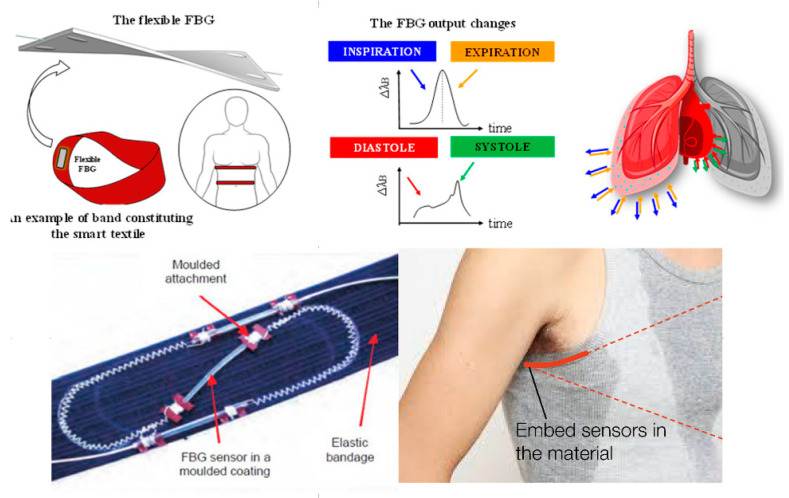
**(Adapted)** An example of smart textile sensors [94,95,96]. A representation of some very novel sensor designs is shown, based on a study to determine the location and configuration of the FBGs in the textile in order to measure the multiple details of breathing and its involved movements. It is intended to show the research that there is for the development of this type of garments that help constant monitoring of vital signs or physical activity.

**Figure 16 biosensors-11-00058-f016:**
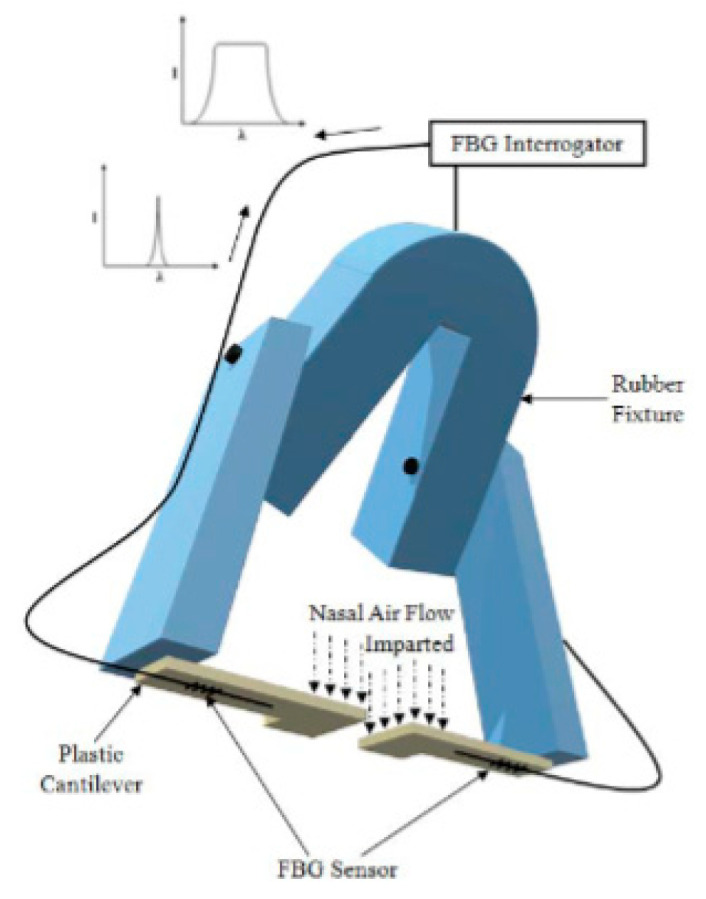
An example of the Pant Shweta prototype, a breathing measurement device. Another design to highlight where, through a very simple prototype, it is possible to measure breathing through a sensor with a FBG for its task. The prototype is most ingenious as it is a nasal bridge that measures the respiratory flow and can measure the respiratory rate from the volume without problems—a totally different and novel proposal. [92].

**Figure 17 biosensors-11-00058-f017:**
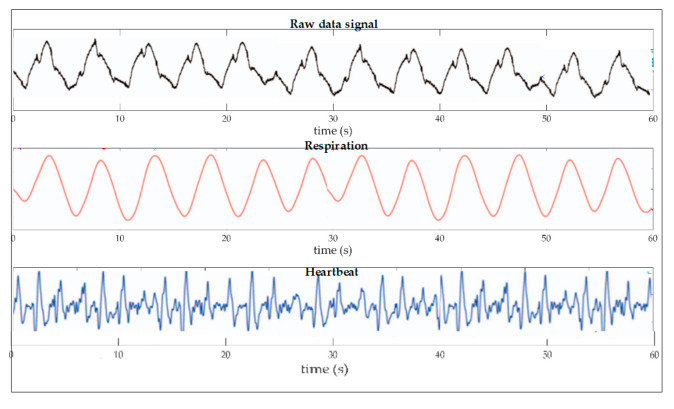
Based on the signals representing the waveforms of the filtered hard data as an example of Bonefacio Julien. It can be seen how the output signal and its respective wavelength appear in black with noise during the analysis performed by the FBG sensor, while the signals in red and blue, BR and HR, respectively, having already been filtered, can be seen in a clearer way due to the elimination of noise—a simple sign that from the same signal can be obtained different data out of phase in time and the feasibility of being able to obtain not only a vital signal to monitor. Here, the complexity lies more in monitoring both in real time [106].

**Figure 18 biosensors-11-00058-f018:**
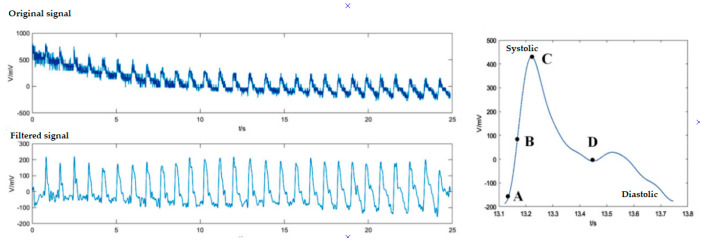
Example of the cardiac pulse flow during monitoring. It shows how the cardiac pulse signal is obtained. At first, background noise is observed in the output signal, as well as thermal and electromagnetic interference, so they proceed to filter the signal and take samples of the pulses to smooth the signal and finally take more samples to process the signal [107].

**Figure 19 biosensors-11-00058-f019:**
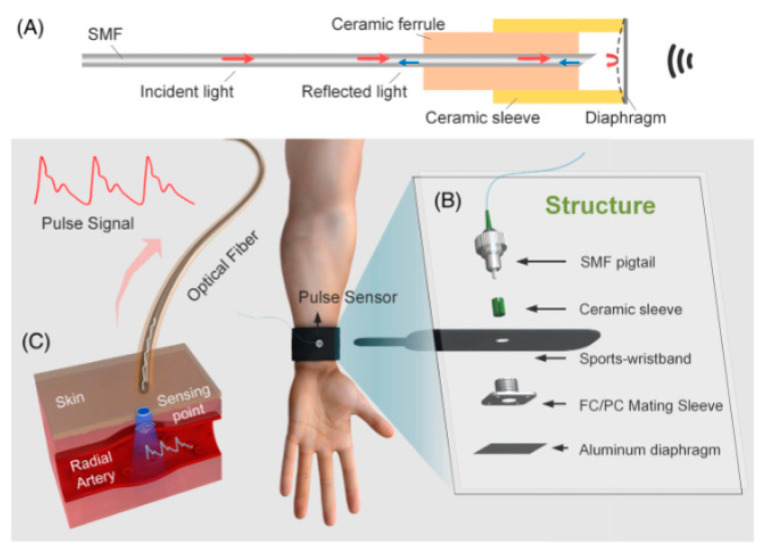
Wang’s figure of an example of a fiber sensor showing its structure, composition, and operation. The model of an OF sensor is shown from its components and how each of them are structured. The whole design for its operation is presented and how the characterized fiber works—together with all its components and operation functions as a wrist blood pressure sensor. A bold design of how vibrations originated by a membrane can be used to obtain readings in the wave [105]. (**A**), Schematicdiagram of optical fiber pulsesensor; (**B**), Structural design ofwearable pulse sensor; (**C**), Thetransmission of pulse waves signals.

**Table 1 biosensors-11-00058-t001:** Body temperature, research with optical fiber sensors.

Fiber Type and Sensor Operating Wavelength (nm)	Sensor Tested on	Sensitivity (nm/°C)	Sensor Technology	Detection Ranges (°C)	Ref.
SMF 1550	Thorax, chest wall	0.31	FBG Encapsulated in PDMS	33 to 37	[6]
SMF 1561.07 to 1561.467	Nose	0.0114	FBG	10 to 44.8	[14]
SMF 1557 and 1627	Heating tube filled with water	0.95 to 1.03	LPFG H and KrF excimer laser	20 to 100	[15]
SMF and Photonic hollow core 1495.4 and 1520.3	Temperature chamber	0.0119 to 0.0138	LPFG Electric arc discharges	30 to 80	[56]
SMF 1515 and 1547	Climatic chamber	0.41066 and 0.40509	LPFG NaOH, KOH, H_2_SO_4_, H_2_O, C_3_H_4_O_2_, NaCl, C_3_H_8_ClN. argon-ion laser (244 nm—UV)	25 to 85	[57]
SMF and Non-core 1588	Thermostatic furnace	-0.00643	LPFG	34 to 154	[58]
MMF 630	distilled water	-9.52519E-05	SPR sensor Au film	30 to 70	[59]
POF 1552.45–1552.65	Chest	−0.055	He–Cd laser for inscription, taped fiber, DPDS dopant and UV irradiation	20 to 50	[60]
SMF 1550	Thorax and neck	-	FBG	35.5 to 39.5	[61]
SMF 1554.1207 1512.5 to 1587.5	Chest	0.010378 to 0.03844	FBG, encapsulated on PDMS, Sylgard 184	35.5 to 37	[62]
1550.218 and 1550.208	Thorax	10.3 to 11.3	FBG, acrylic and fiberglass encapsulated	20 to 90	[63]

SMF: single-mode fiber; MMF: multi-mode fiber; POF: plastic optical fiber; FBG: fiber Bragg grating; PDMS: polydimethylsiloxane; LPFG: long-period fiber grating; H: hydrogen; KrF: krypton fluoride; NaOH: sodium hydroxide; KOH: potassium hydroxide; H_2_SO_4_: sulfuric acid; H_2_O: water; C_3_H_4_O_2_: methylglyoxal; NaCl: sodium chloride; C_3_H_8_ClN: 3-chloropropylamine; SPR: surface plasmon resonance; Au: gold; DPDS: diphenyl disulfide; He-Cd: helium cadmium; UV: ultraviolet.

**Table 2 biosensors-11-00058-t002:** Respiration or breathing rate, research with optical fiber sensors.

Fiber Type and Sensor Operating Wavelength (nm)	Sensor Test Position	Respiratory Rate—Breaths per Minute BR—(bpm)	Sensor Technology	Error or Variation (%)	Ref.
SMF 1550	Thorax, chest wall	13.4 to 19.5 men 14.5 to 19 women	FBG Encapsulated in PDMS	±1.96	[6]
SMF	Tricuspid area and carotid artery	18	MZI, FBG	-	[7]
SMF 1550	Baby upper abdominal area	10 to 100	Curves on the fiber	0.25	[55]
SMF 1510	Thorax, torso, and abdomen	10	LPFG	6 to 8	[56]
SMF 1547.77	Thorax and abdomen (precordium area)	9 to 11	FBG Bending effects OTDR	10	[58]
SMF, hetero-core 1310	Smart textile for the upper abdominal area	16	Macro-bending on the fiber	1	[60]
SMF 1550	Thorax and neck	12 to 24	FBG	2	[61]
SMF 1533 and 1557	Thoraco-abdominal and chest wall surface	6 to 11 on standing, 5 to 11 on supine	FBG	0.38	[62]
SMF 1299 and 1548	Thoraco-abdominal on resuscitation manikin	-	LPFG, FBG	0.4 and 0.8	[76]
SMF 1470.73	Human chest wall on supine	5 to 11 on natural, 6 to 12 on shallow	LPFG	4.4 to 8.7 and 5.8 to 10.1	[78]
SMF 1532 and 1541	Upper thorax	14	FBG	8.3	[81]
MMF 1310 nm	Body back	6 to 14 Average 12.31	Microbendings on the fiber	9.8	[84]

SFM: single-mode fiber; MMF: multi-mode fiber; FBG: fiber Bragg grating; PDMS: polydimethylsiloxane; MZI: Mach–Zehnder interferometer; LPFG: long-period fiber grating; OTDR: optical time-domain reflectometry; BR: breathing rate.

**Table 3 biosensors-11-00058-t003:** Respiration or breathing rate, research with optical fiber sensors with characterization of the nanostructure through doping agents. Characterization with chemicals in the nanostructure.

Fiber Type and Sensor Operating Wavelength (nm)	Sensor Test Position	Respiratory Rate—Breaths per Minute BR—(bpm)	Sensor Technology	Error or Variation (%)	Ref.
SMF 1561.07 to 1561.467	Nose	12 to 18 Average 15	FBG created with H	-	[14]
POF 1552.45–1552.65	Chest	18	He–Cd laser for inscription, taped fiber, DPDS dopant and UV irradiation	-	[60]
SMF 1554.1203 1513.444–1585.787	Thorax	16.22	FBG, PDMS, Sylgard 184	3.9	[62]
SMF 1550.218 and 1550.208	Thorax	16	FBG, acrylic and fiberglass for protection	4.64	[63]
SMF, MMF, TMF	–	18	MZI		[69]
SMF 1440 to 1550, best on 1519.95	Chest of respiratory manikin	–	LPFG, FBG UV and argon-ion laser induced Er doped and core of GeO_2_/SiO_2_, inner cladding of SiO_2_, outer cladding of SiO_2_/F/P_2_O_5_	-	[79]
SMF 1533 to 1553	Chest wall	11 to 12	FBG polymeric glue and POF	0.3	[83]
SMF	Chest	Average of 17	FBG, Ge doped fiber		[86]
SMF 1554.1204 nm	Chest	15.4925 standing, 15.5119 supine, 15.7638 sitting	FBG PDMS for encapsulation, Sylgard 184, CH_3_Cl as dopant	4.4	[89]
MMF, ECF 1550	Back of the body seated	16 to 20	FBG Microbending, He-Ne laser, mechanical induction	15	[90]
SMF 1550	Back of the body on supine	14 to 19	FBG, PDMS	4.41	[91]
SMF 1536 and 1548	Nose bridge	9.64 to 10.76	FBG Ge doped, strain variations	1.3	[92]
SMF 1554.1207 nm	Chest	21.6676 standing 20.6386 seated 14.8741 back	FBG, encapsulated on PDMS, Sylgard 184	-	[93]

SFM: single-mode fiber; MMF: multi-mode fiber; TMF: two-mode fiber; POF: plastic optical fiber; FBG: fiber Bragg grating; MZI: Mach–Zehnder interferometer; Ge: germanium; H: hydrogen; Er: erbium; PDMS: polydimethylsiloxane; LPFG: long-period fiber grating; ECF: eccentric core fiber; DPDS: diphenyl disulfide; UV: ultraviolet. GeO_2_: germanium dioxide; SiO_2_: silicon dioxide; F: fluorine; P_2_O_5_: phosphorus pentoxide. CH_3_Cl: chloromethane; He–Ne: helium neon; He–Cd: helium cadmium.

**Table 4 biosensors-11-00058-t004:** Pulse or heart rate, research with optical fiber sensors.

Fiber Type and Sensor Operating Wavelength (nm)	Sensor Test Position	Heart Rate—Beats per Minute HR—(BPM)	Sensor Technology	Error or Variation (%)	Ref.
SMF 1550	Thorax, chest wall	64 to 81 men 67 to 98 women	FBG Encapsulated in PDMS	±1.96	[6]
SMF	Tricuspid area and carotid artery	61	MZI, FBG	-	[7]
SMF 1550	Chest	57.5	FBG Encapsulated in PDMS	1.96	[12]
POF 1552.45 - 1552.65	Chest	150	He–Cd laser for inscription, taped fiber, DPDS dopant and UV irradiation	±2	[60]
SMF 1550	Thorax and neck	60 to 120	FBG	2	[61]
SMF 1554.1203 1513.444–1585.787	Thorax	78.54	FBG, encapsulated on PDMS, Sylgard 184	1.96	[62]
SMF 1550.218 and 1550.208	Thorax	74.3	FBG, acrylic and fiberglass encapsulated	4.87	[63]
SMF, MMF, TMF	-	66	MZI	-	[69]
MMF 1310 nm	Body back	77 to 83 Average 66.55	Microbendings on the fiber	0.6	[84]
SMF	Chest	Average of 107	FBG, Ge doped	-	[86]
SMF 1545 to 1555	Radial artery at the wrist	51	MZI, FBG, InGaAs	-	[87]
SMF 1554.1207 nm	Chest	62,8363 standing 61,9159 seated 76,8499 back	FBG, PDMS, Sylgard 184	-	[93]
MMF	Back of the head	58 to 74	Microbendings on fiber	2.7 to 3.44	[100]
SMF 1538.4 to 1538.6	Back of the body	76.8	FBG	< 7.4	[101]
SMF 1549.5 to 1550.5	Temple (best), finger, ankle (worst) and dorsum pedis	Average of 66.33, 60.33, 60 and 57.66	FBG, MZI	1.47 (best) 28.33 (worst)	[102]
MMF	Back of the body	84	Mechanical induction	7.31	[103]
POF 950	Neck and chest	68 (best) and 52 (worst)	PDMS and plastic polymer macro-bending and strain	-	[104]
SMF 1550 nm	Wrist	66	Er 12µm thick Al diaphragm	5	[105]

SFM: single-mode fiber; MMF: multi-mode fiber; TMF: two-mode fiber; POF: plastic optical fiber; FBG: fiber Bragg grating; MZI: Mach–Zehnder interferometer; Ge: germanium; Er: erbium; PDMS: polydimethylsiloxane; ECF: eccentric core fiber; DPDS: diphenyl disulfide; UV: ultraviolet; He–Cd: helium cadmium; InGaAs: indium gallium arsenide; Al: aluminum.

**Table 5 biosensors-11-00058-t005:** Blood pressure, research with optical fiber sensors.

Fiber Type and Sensor Operating Wavelength (nm)	Sensor Test Position	Blood Pressure (mmHg)	Sensor Technology	Error or Variation (mmHg)	Ref.
SMF 1549.5 to 1550.5	Temple (best), finger, ankle(worst) and dorsum pedis	131/73.5 average	MZI, FBG	± 3	[102]
SMF 1550	Wrist	116.5/71.75 average	Er, Al diaphragm	-	[105]
SMF 550 to 700	On a goat left ventricle, left atrium, right atrium, and aorta	–100 to 400	FPI, PI, C_8_H_20_O_4_Si	± 4	[110]
SMF, MMF 1547.5	In-vivo coronary artery of a swine	54 to 88 in aortic arch 60 to 100 in right coronary artery	FPI cavity with HF, SiO_2_ diaphragm	-	[111]
SMF 1549.5 to 1550.5 1559.5 to 1560.5	**N**eck and ankle	106 to 119 and 109 to 122	MZI, FBG	7 and 5 on systolic	[112]
SMF, MMF	Tortuous vessels of a swine model in-vivo	54 to 88	FPI, stretched core MMF, SiO_2_ diaphragm	-	[113]
SMF 1549.5 to 1550.5	Right wrist	110.7 supine 107.3 sitting 103.8 standing	MZI, FBG, SiO_2_, InGaAs	3 supine 2.8 sitting 3.8 standing	[114]
SMF 1525 to 1575	Wrists	123.6 average with no cover 119.2 average with cover	FBG	2 no covered 6 covered	[115]
POF 15543 and 1553	Left arm	106.5/65 average	FBG, glue NORLAND 78	5/3	[116]

SFM: single-mode fiber; MMF: multi-mode fiber; POF: plastic optical fiber; FBG: fiber Bragg grating; MZI: Mach–Zehnder interferometer; FPI: Fabry–Pérot interferometer; PI: polyimide, C_8_H_20_O_4_Si: tetraethoxysilane; HF: hydrofluoric acid; SiO_2_: silicone dioxide; Er: erbium; InGaAs: indium gallium arsenide; Al: aluminum.

**Table 6 biosensors-11-00058-t006:** Multiparametric—vital signs, research on optical fiber sensors.

Number of Sensors	Body Temperature	Respiration or Breathing Rate	Pulse or Heart Rate	Blood Pressure	Ref.
3	**×**	**×**	**×**		[6]
2		**×**	**×**		[7]
1	**×**	**×**			[14]
2	**×**	**×**	**×**		[60]
2	**×**	**×**	**×**		[61]
2	**×**	**×**	**×**		[62]
1	**×**	**×**	**×**		[63]
2		**×**	**×**		[69]
1		**×**	**×**		[84]
1		**×**	**×**		[86]
1		**×**	**×**		[93]
1			**×**	**×**	[105]

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
