# Peer review of "Fiber Optic Sensors for Vital Signs Monitoring. A Review of Its Practicality in the Health Field"

_biosensors, 2021, doi:10.3390/bios11020058_

Round 1
Reviewer 1 Report
In this work, the authors propose a review of fiber optic sensors for the monitoring of vital signs, mainly temperature, respiratory rate and heart rate. The sensing structure considered are mostly based on fiber Bragg gratings (FBG), long period gratings (LPG) and interferometers.
I cannot recommend the publication of the paper in the current form due to the following reasons:
The use of English language should be greatly improved in the whole manuscript, the authors should carefully address this aspect.
The authors commented a relatively long list of works but it seems that a comparison between them is not clearly reported. As a review paper we expect not only a descriptive summary of the topic, but also some critical discussion providing any synthesis, connection, or critique.
Additional comments:
Eq. (1), n is the effective refractive index of the core mode.
Additional theoretical background should be reported regarding interferometers.
Author Response
The use of English language should be greatly improved in the whole manuscript, the authors should carefully address this aspect.
It will be come again to the translation in English language of all the document once taken care of all the other observations.
The authors commented a relatively long list of works but it seems that a comparison between them is not clearly reported. As a review paper we expect not only a descriptive summary of the topic, but also some critical discussion providing any synthesis, connection, or critique.
Unlike the systems of traditional detection and monitoring of vital signs, with the optical fiber sensors they have very attractive characteristics like excellent sensitivity, great rank of operation and a high trustworthiness. The advantage of the optical fiber use is that they are elaborated with dielectric materials that are chemically inert, see-compatible and immune to the electromagnetic interference and many of them they can be covered by designed materials to support great temperatures; these properties make them excellent so that in his characterization as sensors, they can be adapted in hostile surroundings and aggressive like the interior of some block of construction, materials or structures, a system of generation and transmission of electricity, where the conventional sensors can have an unstable operation and by the same it increases the possibility of faults [1]. According to the data that appear in the different tables it can deduce that the measurement or monitoring of the vital signs varies depending to the atmosphere where the registries are made, in very few of the obtained results is registered that they have left the analysis of a laboratory, reason why is recommended that the tests of carry out in real environment and different regions, it is desired that some designs works of investigation can take to stay as prototypes. Comparing the presented results of the different works and articles from investigation, the ranks of reached operation and sensitivity are useful for particular applications where each equipment of investigation establishes its guidelines like units of measurement, operation limits, this makes difficult to be able to establish with certainty the feasibility of its operation outside the research laboratories, separate that single mobility and handling are considered under certain very specific conditions of.
The results obtained in some works show an enormous variation solely with changing the area of positioning of the sensor, it is the case of Chinese, S. Et al. [2] where the samples are obtained from four parts of the body (temple, finger, ankle and dorsum pedis) for the obtaining of the pulse cardiac. This puts in evidence that, if they wanted to obtain good results for the obtaining of some vital signs, is necessary to study medical Literature on the different vital signs like the physiology, that firstly allows to establish parameters for the correct taking of samples and tests.
In addition, in some cases they show to similar levels of sensitivity used fibers of high cost in some articles, that is to say, an alternative of smaller cost is appearing, but even so he is not sufficiently attractive like even competing with the devices sensing traditional vital signs [3-5] that every time seem to be more precise and reliable. Opposite case, very good works are had whose results were not really expected [6] since its percentage of error or variation was very high, due to factors like temperature differences or pressure, nevertheless continues being good studies since they indicate methodologies to follow or to correct for future projects.
Additional comments:
Eq. (1), n is the effective refractive index of the core mode.
Correction in line 93 of the document:
Where the wavelength is , n is the effective refractive index of the core mode.
Additional theoretical background should be reported regarding interferometers.
Continuing with the operation principles we can analyze the utility of the interferometers as devices whose operation is in the effect caused by the interference of two beams of light, which propagate by different optical ways through one or more optical fibers [7]. It is this form of operation in where a light beam separated or is grouped that is required altogether of these modules for its work with optical fibers; we can basically make emphasis in four types of interferometers, whose configurations are those of Fabry-Pérot, Mach-Zehnder, Michelson and Sagnac which they widely have been used and demonstrated [8] and the main characteristics and differences among them in their application like optical fiber sensors will be described.
Used for extremely precise measurements in very small distances, its design also is known like “etalon”. Its elaboration consists to each other of two flat, parallel semitransparent mirrors by a fixed distance, where a light wave that affects the surface with some reflected arbitrary angle will experience manifold within the adjustment of mirrors as in Figure 1. In optical fibers, one was that it is possible to be created through a ring (gyroscope) that works as guide of wave, this is possible if the optical fiber conserves a defined front of affluent wave and if the effectiveness in the connection of the light in the fiber is not very small[ 9].
Figure 1 - Sample of the operation of the Fabry-Pérot interferometer
Here the form to operate of the Fabry-Pérot interferometer can be seen, where in the parallel surfaces the light beam is reflected multiple times where interference is originated [10]
Optical fiber Mach-Zehnder interferometer:
Its structure consists of two couplers which divide the optical power of entrance in two equal parts, thus the light travels through two different ways, a way operating like reference and the other is used for the sensing one, being a desirable characteristic that the optical fibers have the same length, so that the resistance in the interference pattern is so great at the most is equal this length. In figure 2 (a) it is possible to observed since the beam of incident light is divided in two arms by means of a coupler later to recombine itself by means of a second coupler, the recombined beam of light has the interference component depending on the optical way between the arms. When it happens the detection the reference arm stays isolated of any external variation and single the sensor arm is exposed to the changes. These changes in the sensor arm are what they originate the difference of optical way with which it is possible to make an analysis of the variation in the signal of interference for later, the interference pattern is detected by the photodetectors so that one measures the signal and the other solely reference [11].
Optical fiber Michelson interferometer:
In this optical fiber sensor that is present in figure 2 (b), the coupler splits the beam of light in two different optical ways where, the light reflected by the mirrors recombines by means of the coupler giving origin to an interference pattern that arrives for its analysis at the photodetector, can be said that its way of operation is very similar to the Mach-Zehnder interferometers, in fact a Michelson is like half of Mach-Zehnder as far as its configuration since single uses a coupler and a photodetector; basically the interference takes place between the beams of both arms, but each one of the light beams is reflected at the end of each arm in an Michelson interferometer. The main difference between these two interferometers is the reflectors reason why their manufacture and principle of operation are almost the equal ones. In this device Faraday rotator mirrors are used to maintain the polarization of the separated beams in the fiber [12].
Optical fiber Sagnac interferometer:
In this type of interferometer that is in figure 2(c), the light beam enters the coupler by one of entrance fibers as it is possible to be observed in the same figure, giving rise to that the light divides in two beams with the same intensity and each one of them moving in opposed directions through the optical fiber, in the end is going to be the side of the configuration of Sagnac the one that goes towards the photodetector. This phenomenon it is known like detection of rotation, which is obtained placing the device in a spin table and is when these turns happen that the lines of the interference pattern are moved [13].
Optical fiber Fabry-Pérot interferometer:
It is considered simplest since its single configuration it is needed a circulador and a photodetector; for this the interference plays a fundamental role since this is originated by means of a cavity in some of the ends of the optical fiber as is in the figure 2(d), where a light beam that is originated in the source travels through the fiber towards the circulador, and when arriving at the cavity the interference for later takes place, to return towards the photodetector. The cavity of Fabry-Pérot is made up of two separated parallel reflecting surfaces; the interference happens due to the multiple superpositions of the reflected and transmitted beams of light in these parallel surfaces [14].
Figure 2 - Configuration of different optical fiber interferometers.
(a) The figure exemplifies the basic operation of a fiber Mach-Zehnder interferometer type, from the exit of the beam of the light source to its analysis through the photodetectors. (b) It shows a sensor using a fiber Michelson interferometer, which if it is analyzed is very similar in configuration to Mach-Zehnder, safe with less the rotatory devices and mirrors. (c) It shows the configuration of the fiber Sagnac interferometer where the opposite ways are observed with clarity that the beam takes from light. (d) It shows a fiber Fabry-Pérot interferometer type and the way in which the light beam travels from the source to its return to photodetector [1].
In figure 3 are the different configurations from as much extrinsic as intrinsic Fabry-Pérot fiber sensors type, that is important since it shows to us with greater detail and clarity the form in which the optical fiber can be characterized so that it works like a sensor, is possible to indicate that each interferometer has its different configurations and that some are very ingenious variations that have been adapted according to the type of means to be registered since one better response is obtained.
Figure 3 - Configurations of Fabry-Pérot fiber sensors.
The figure shows the intrinsic and extrinsic configurations of Fabry-Pérot fiber sensors the where: (a) It represents a cavity formed by an internal mirror at the end of a fiber; (b) a cavity formed by two internal mirrors; (c) a cavity formed by two fiber Bragg gratings. For each case L represents the length of the optical cavity. (d) represents a cavity formed by a diaphragm at the end of a fiber; (e) a cavity formed by the surfaces of a cover in the end of the fiber; (f) a cavity formed in the end of a singlemode fiber and an aligned multimode fiber through a capillary; (g) a cavity formed by a singlemode fiber that the end is joined with a hollowcore fiber. And the equal one in each case L represents the length of the optical cavity, which is an air bubble in the configurations (d), (f), and (g) [15].
In figure 4 some of the most common configurations appear on the fiber Mach-Zehnder sensors type which take place in changes in micro structure of the fiber by taper, joints with other fibers, displacements of the cores between one or more fibers or with multicore fibers, where the fundamental is to make the difference of optical ways through the interference.
Figure 4 - Different configurations from Mach-Zenher fiber sensor.
Here are different fiber Mach-Zehnder interferometers where we can stand out: (a) shows an interferometer created through taper the optical fiber, same that is obtained applying force in opposite directions when a unloading by electrical arc is made to modify the nano structure of optical fiber[16]; (b) shows joints of optical fiber but with a space displacement in the cores, which originates interference[17]; (c) shows joints of several types of optical fiber where in first a fiber Bragg grating type created on a single mode fiber that goes joined with an optical fiber with four cores that as well hold a concealed knife with another single mode fiber segment [ 18]. They are different techniques that are used to generate the sensors and like the configurations of Fabry-Pérot, these are made to obtain greater sensitivity, specificity, stability or exactitude in the sensor that is wanted to create.
REFERENCES
- Wang, L.; Fang, N. Applications of Fiber-Optic Interferometry Technology in Sensor Fields. In Optical Interferometry; Banishev, A.A., Bhowmick, M., Wang, J., Eds.; InTech, 2017 ISBN 978-953-51-2955-4.
- Chino, S.; Ishizawa, H.; Hosoya, S.; Koyama, S.; Fujimoto, K.; Kawamura, T. Research for Wearable Multiple Vital Sign Sensor Using Fiber Bragg Grating -Verification of Several Pulsate Points in Human Body Surface.; May 1 2017.
- Dass, S.; Jha, R. Microfiber-Wrapped Bi-Conical-Tapered SMF for Curvature Sensing. IEEE Sens. J. 2016, 16, 3649–3652, doi:10.1109/JSEN.2016.2531748.
- Dash, J.N.; Dass, S.; Jha, R. Photonic Crystal Fiber Microcavity Based Bend and Temperature Sensor Using Micro Fiber. Sens. Actuators Phys. 2016, 244, 24–29, doi:10.1016/j.sna.2016.04.016.
- Dass, S.; Jha, R. Micron Wire Assisted Inline Mach-Zehnder Interferometric Curvature Sensor. IEEE Photonics Technol. Lett. 2015, 28, 1–1, doi:10.1109/LPT.2015.2478957.
- Hu, H.; Sun, S.; Lv, R.; Zhao, Y. Design and Experiment of an Optical Fiber Micro Bend Sensor for Respiration Monitoring. Sens. Actuators Phys. 2016, 251, 126–133, doi:10.1016/j.sna.2016.10.013.
- Lee, B.H.; Kim, Y.H.; Park, K.S.; Eom, J.B.; Kim, M.J.; Rho, B.S.; Choi, H.Y. Interferometric Fiber Optic Sensors. Sensors 2012, 12, 2467–2486, doi:10.3390/s120302467.
- Zhu, T.; Wu, D.; Liu, M.; Duan, D.-W. In-Line Fiber Optic Interferometric Sensors in Single-Mode Fibers. Sensors 2012, 12, 10430–10449, doi:10.3390/s120810430.
- Vali, V.; Shorthill, R.W. Fiber Ring Interferometer. Appl. Opt. 1976, 15, 1099, doi:10.1364/AO.15.001099.
- Silvestri, S.; Sche, E. Optical-Fiber Measurement Systems for Medical Applications. In Optoelectronics - Devices and Applications; Predeep, P., Ed.; InTech, 2011 ISBN 978-953-307-576-1.
- Heideman, R.G.; Lambeck, P.V. Remote Opto-Chemical Sensing with Extreme Sensitivity: Design, Fabrication and Performance of a Pigtailed Integrated Optical Phase-Modulated Mach–Zehnder Interferometer System. Sens. Actuators B Chem. 1999, 61, 100–127, doi:10.1016/S0925-4005(99)00283-X.
- Kashyap, R.; Nayar, B. An All Single-Mode Fiber Michelson Interferometer Sensor. J. Light. Technol. 1983, 1, 619–624, doi:10.1109/JLT.1983.1072153.
- Starodumov, A.N.; Zenteno, L.A.; Monzon, D.; De La Rosa, E. Fiber Sagnac Interferometer Temperature Sensor. Appl. Phys. Lett. 1997, 70, 19–21, doi:10.1063/1.119290.
- Hernández, D.D.M. Ing. Fís. Carmen Edith Domínguez Flores. 98.
- Yin, S.; Ruffin, P.B.; Yu, F.T.S. Fiber Optic Sensors; CRC Press, 2017; ISBN 978-1-4200-5366-1.
- Aref, S.H. Physical Measurement with In-Line Fiber Mach-Zehnder Interferometer Using Differential Phase White Light Interferometry. Opt. Fiber Technol. 2017, 38, 98–103, doi:10.1016/j.yofte.2017.09.001.
- Tian, Z.; Yam, S.S.-H.; Loock, H.-P. Single-Mode Fiber Refractive Index Sensor Based on Core-Offset Attenuators. IEEE Photonics Technol. Lett. 2008, 20, 1387–1389, doi:10.1109/LPT.2008.926832.
- Yu, J.; Xu, S.; Jiang, Y.; Chen, H.; Feng, W. Multi-Parameter Sensor Based on the Fiber Bragg Grating Combined with Triangular-Lattice Four-Core Fiber. Optik 2020, 208, 164094, doi:10.1016/j.ijleo.2019.164094.

Reviewer 2 Report
This manuscript presents a short review of the technological advancements in fiber optics for Vital signs monitoring. This manuscript is very interesting and is undoubtedly a crucial subject, and therefore is an excellent addition to the literature. However, the manuscript here presented is poorly executed and requires extensive proofreading and re-organization before publication.
The biosensor definition (line 47) in the introduction is confusing and should be clarified. A biosensor is an analytical device used to detect a chemical substance that combines a biological motif with a physicochemical detector. It seems that most of the sensors described in this manuscript do not use biological motifs coupled with the optical fibers. Introducing the biosensor subject seems pointless since these are not considered biosensors at all. Also, the last paragraph of the introduction is too long and should be simplified.
In most of the tables presented in this manuscript, different units are described for the same measurements; these values should be uniformed to facilitate any comparison.
Tables of contents and abbreviations should be added to the manuscript. The abbreviations used throughout the text should be uniformized. For instance, in the introduction, the authors have consistently used the extended term "optical fiber". In section 2, suddenly they change for the abbreviation OF. Also, in many situations, the authors use semicolons, where it should be used periods.
The general mechanisms of detection should be illustrated and explained briefly instead of constant repetition throughout the text. For instance, the major mechanism of detection for breathing and/or respiratory rate seems to be by sampling the human body's chest and abdomen movement. Afterwards, only the sensor characteristics should be discussed by the authors. This should be extended to the remaining sections if possible.
There are two sections 2.2.
Reference 48 should be discussed in section 2.3.
A comparison (advantage and disadvantages) of fiber optics-based methods with current technologies used in medicine should be discussed.
The authors should present a critical conclusion about the field and how this field should evolve in the upcoming years to surpass the current drawbacks and become a capable technology for medicine consideration.
Thank you
Author Response
The biosensor definition (line 47) in the introduction is confusing and should be clarified. A biosensor is an analytical device used to detect a chemical substance that combines a biological motif with a physicochemical detector. It seems that most of the sensors described in this manuscript do not use biological motifs coupled with the optical fibers. Introducing the biosensor subject seems pointless since these are not considered biosensors at all. Also, the last paragraph of the introduction is too long and should be simplified.
Taking into consideration the comments on the changes in the introduction, the introduction was modified as follows
The penetration of fiber optic sensors in the medical or healthcare market is very low or almost null in home devices due to their high cost or too many regulations, which hinder their entry, however, there have been small advances as these sensors are used in high precision surgery or in magnetic resonance imaging, their use is increasingly used in various technologies or ancillary equipment, so it has advanced one step at a time [1].
Since the 1980s [2], optical fiber sensors have been used for real-time pressure measurement of tendons [3] and thereafter in a wide variety of possible medical applications as pressure pads in contact with the skin [4], cardiovascular or even invasive sensors during urodynamic analysis [5]. Many researches have focused their studies on monitoring and detection of vital signs through fiber optic sensors that are located in some medium in contact with the skin [6] with the capacity to be composed of two or more sensors on the same fiber [7], placed in textile vestments [8,9] very novel in their shape, an ingenious way to install them at the same time that they provide protection to the optical fiber [10], because vital signs are the first way to test a person's health and stability, with heart pulse monitoring being a well-studied field [11] but one that lends itself to continuing to innovate and develop improvements in measurement devices and equipment [12,13].
Fiber optic sensors have been used in many applications for the measurement of chemical parameters, liquid flow and levels, gas detection, but mainly in the electrical and mechanical fields due to the great advantages that have been attributed to them such as immunity to electromagnetic interference, apart from the small size of the fiber that makes them perfect for the development of lightweight and mechanically robust sensors [14]. However, in the medical field they have not yet gained wide acceptance since conventional sensors have also made great advances, giving them good user characteristics such as reliability, maintenance and support, and technological integration [15]. There is a great need for a simple system for measuring vital signs for home health monitoring.
In most of the tables presented in this manuscript, different units are described for the same measurements; these values should be uniformed to facilitate any comparison.
The units of the tables were unified and the information was distributed in a better way to present and highlight the differences, characteristics and operation of the sensors, as follows we present how they were updated:
Table 1 – Body temperature, works with optical fiber sensors.
|
Fiber type and sensor operating wavelength (nm) |
Sensor tested on |
Sensitivity (nm/°C) |
Sensor technology |
Detection Ranges (°C) |
Ref. |
|
SMF 1550 |
Thorax, chest wall |
0.31 |
FBG Encapsulated in PDMS |
33 to 37 |
[6]8 |
|
SMF 1561.07 to 1561.467 |
Nose |
0.0114 |
FBG |
10 to 44.8 |
[16]33 |
|
SMF 1557 and 1627 |
Heating tube filled with water |
0.95 to 1.03 |
LPFG H and KrF excimer laser |
20 to 100 |
[17]38 |
|
SMF and Photonic hollow core 1495.4 and 1520.3 |
Temperature chamber |
0.0119 to 0.0138 |
LPFG Electric arc discharges |
30 to 80 |
[18]39 |
|
SMF 1515 and 1547 |
Climatic chamber |
0.41066 and 0.40509 |
LPFG NaOH, KOH, Hâ‚‚SOâ‚„, H2O, C3H4O2, NaCl, C3H8ClN. argon-ion laser (244 nm - UV) |
25 to 85 |
[19]40 |
|
SMF and Non-core 1588 |
Thermostatic furnace |
-0.00643 |
LPFG, Multiple fluids |
34 to 154 |
[20]41 |
|
MMF 630 |
distilled water |
-9.52519E-05 |
Surface plasmon resonance (SPR) sensor Au film |
30 to 70 |
[21]42 |
|
Plastic optical fiber (POF) 1552.45 – 1552.65 |
Chest |
−0.055 |
He-Cd laser for inscription, taped fiber, diphenyl disulphide (DPDS) dopant and UV irradiation |
20 to 50 |
[22]79 |
|
SMF 1550 |
Thorax and neck |
- |
FBG |
35.5 to 39.5 |
[23]91 |
|
SMF 1554.1207 1512.5 to 1587.5 |
Chest |
0.010378 to 0.03844 |
FBG, encapsulated on Polydimethylsiloxane, Sylgard 184 |
35.5 to 37 |
[24]94 |
|
1550.218 and 1550.208 |
Thorax |
10.3 to 11.3 |
FBG, acrylic and fiberglass encapsulated |
20 to 90 |
[25]95 |
Table 2 - Respiration or respiratory rate, works with optical fiber sensors.
|
Fiber type and sensor operating wavelength (nm) |
Sensor test position |
Respiratory Rate - breaths per minute RR - (bpm) |
Sensor techonoly |
Error or variation (%) |
|
Ref. |
|
SMF 1550 |
Thorax, chest wall |
13.4 to 19.5 men 14.5 to 19 women |
FBG Encapsulated in PDMS |
±1.96 |
|
[6]8 |
|
SMF |
Tricuspid area and carotid artery |
18 |
MZI, FBG
|
- |
|
[7]9 |
|
SMF 1470.73 |
Human chest wall on supine |
5 to 11 on natural, 6 to 12 on shallow |
LPFG |
4.4 to 8.7 and 5.8 to 10.1 |
|
[26]19 |
|
SMF 1550 |
Baby upper abdominal area |
10 to 100 |
Curves on the fiber |
0.25 |
|
[55]55 |
|
SMF 1510 |
Thorax, torso and abdomen |
10 |
LPFG |
6 to 8 |
|
[56]56 |
|
SMF 1299 and 1548 |
Thoraco-abdominal on resuscitation manikin |
- |
LPFG, FBG |
0.4 and 0.8 |
|
[29]57 |
|
SMF 1547.77 |
Thorax and abdomen (precordium area) |
9 to 11 |
FBG Bending effects optical time-domain reflectometry (OTDR) |
10 |
|
[58]58 |
|
SMF, hetero-core 1310 |
Smart textile for de upper abdominal area |
16 |
Macro bending on the fiber |
1 |
|
[60]60 |
|
SMF 1532 and 1541 |
Upper thorax |
14 |
FBG |
8.3 |
|
[61]61 |
|
SMF 1533 and 1557 |
Thoraco-abdominal and chest wall surface |
6 to 11 on standing, 5 to 11 on supine |
FBG |
0.38 |
|
[62]62 |
|
SMF 1550 |
Thorax and neck |
12 to 24 |
FBG |
2 |
|
[23]91 |
|
MMF 1310 nm |
Body back |
6 to 14 Average 12.31 |
Microbendings on the fiber |
9.8 |
|
[34]92 |
Table 3 – Respiration o breathing rate, works with optical fiber sensors with characterization of the nanostructure through doping agents.
|
Fiber type and sensor operating wavelength (nm) |
Sensor test position |
Respiratory Rate - breaths per minute RR - (bpm) |
Sensor techonology |
Error or variation (%) |
|
Ref. |
|
SMF |
Chest |
Average of 17 |
FBG, germanium doped fiber |
|
|
[35]18 |
|
SMF 1561.07 to 1561.467 |
Nose |
12 to 18 Average 15 |
FBG created with hydrogen |
- |
|
[16]33 |
|
SMF, MMF and two mode fiber (TMF) |
- |
18 |
MZI |
|
|
[36]51 |
|
SMF 1440 to 1550, best on 1519.95 |
Chest of respiratory manikin |
- |
LPFG, FBG UV and argon-ion laser induced Erbium doped and core of GeO2/SiO2, inner cladding ofSiO2, outer cladding ofSiO2/F/P2O5 hydrogenation at 120 bar |
- |
|
[37]59 |
|
SMF 1533 to 1553 |
Chest wall |
11 to 12 |
FBG polymeric glue and POF |
0.3 |
|
[38]63 |
|
SMF 1554.1204 nm |
Chest |
15.4925 standing, 15.5119 supine, 15.7638 sitting |
FBG Polydimethylsiloxane for encapsulation, Sylgard 184 and CH3Cl as dopants |
4.4 |
|
[39]66 |
|
MMF 1550 |
Back of the body seated |
16 to 20 |
FBG, eccentric core fiber (ECF) Micro bending, He-Ne laser, mechanical induction |
15 |
|
[40]67 |
|
SMF 1550 |
Back of the body on supine |
14 to 19 |
FBG, polydimethylsiloxane polymer (PDMS) |
4.41 |
|
[41]68 |
|
SMF 1536 and 1548 |
Nose bridge |
9.64 to 10.76 |
FBG Germanium doped fiber, strain variations |
1.3 |
|
[42]69 |
|
POF 1552.45 - 1552.65 |
Chest |
18 |
He-Cd laser for inscription, taped fiber, diphenyl disulphide (DPDS) dopant and UV irradiation |
- |
|
[22]79 |
|
SMF 1554.1207 nm |
Chest |
21.6676 standing 20.6386 seated 14.8741 back |
FBG, encapsulated on Polydimethylsiloxane, Sylgard 184 |
- |
|
[43]93 |
|
SMF 1554.1203 1513.444 - 1585.787 |
Thorax |
16.22 |
FBG, encapsulated on Polydimethylsiloxane, Sylgard 184 |
3.9 |
|
[24]94 |
|
SMF 1550.218 and 1550.208 |
Thorax |
16 |
FBG, acrylic and fiberglass for protection |
4.64 |
|
[25]95 |
Table 4 - Pulse or heart rate, works with optical fiber sensors.
|
Fiber type and sensor operating wavelength (nm) |
Sensor test position |
Heart Rate - Beats per minute HR - (Bpm) |
Sensor technology |
Error or variation (%) |
Ref. |
|
SMF 1550 |
Thorax, chest wall |
64 to 81 men 67 to 98 women |
FBG Encapsulated in PDMS |
±1.96 |
[6]8 |
|
SMF |
Tricuspid area and carotid artery |
61 |
MZI, FBG
|
- |
[7]9 |
|
SMF 1550 |
Chest |
57.5 |
FBG Encapsulated in PDMS |
1.96 |
[12]14 |
|
SMF |
Chest
|
Average of 107 |
FBG, germanium doped fiber |
- |
[35]18 |
|
SMF, MMF and two mode fiber (TMF) |
- |
66 |
MZI |
- |
[36]51 |
|
SMF 1545 to 1555 |
radial artery at the wrist |
51 |
Mach-Zehnder interferometer (MZI), FBG, InGaAs |
- |
[44]64 |
|
MMF |
Back of the head |
58 to 74 |
Microbendings on fiber |
2.7 to 3.44 |
[45]74 |
|
SMF 1538.4 to 1538.6 |
Back of the body |
76.8 |
FBG |
< 7.4 |
[46]75 |
|
SMF 1549.5 to 1550.5 |
Temple (best), finger, ankle(worst) and dorsum pedis |
Average of 66.33, 60.33, 60 and 57.66 |
FBG, MZI |
1.47 (best) 28.33 (worst) |
[47]76 |
|
MMF |
Back of the body |
84 |
Mechanical induction |
7.31 |
[48]77 |
|
POF (plastic optical fiber) 950 |
Neck and chest |
68 (best) and 52 (worst) |
Polymethylmethacrylate and plastic polymer macro-bending and strain |
- |
[49]78 |
|
POF 1552.45 - 1552.65 |
Chest |
150 |
He-Cd laser for inscription, taped fiber, diphenyl disulphide (DPDS) dopant and UV irradiation |
± 2 |
[22]79 |
|
SMF 1550 nm |
Wrist |
66 |
Erbium 12µm thick aluminum diaphragm |
5 |
[50]88 |
|
SMF 1550 |
Thorax and neck |
60 to 120 |
FBG |
2 |
[23]91 |
|
MMF 1310 nm |
Body back |
77 to 83 Average 66.55 |
Microbendings on the fiber |
0.6 |
[34]92 |
|
SMF 1554.1207 nm |
Chest |
62,8363 standing 61,9159 seated 76,8499 back |
FBG, encapsulated on Polydimethylsiloxane, Sylgard 184 |
- |
[43]93 |
|
SMF 1554.1203 1513.444 - 1585.787 |
Thorax |
78.54 |
FBG, encapsulated on Polydimethylsiloxane, Sylgard 184 |
1.96 |
[24]94 |
|
SMF 1550.218 and 1550.208 |
Thorax |
74.3 |
FBG, acrylic and fiberglass encapsulated
|
4.87 |
[25]95 |
Table 5 - Blood pressure, research with optical fiber sensors.
|
Fiber type and sensor operating wavelength (nm) |
Sensor Test position |
Blood pressure (mmHg) |
Sensor technology |
Error or variation (mmHg) |
Ref. |
|
SMF 1549.5 to 1550.5 |
Temple (best), finger, ankle(worst) and dorsum pedis |
131/73.5 average |
MZI, FBG |
± 3 |
[47]76 |
|
SMF 550 to 700 |
On a goat left ventricle, left atrium, right atrium and aorta |
-100 to 400 |
Fabry-Pérot Interferometer (FPI), polyimidelayer, tetraethoxysilane |
± 4 |
[51]82 |
|
SMF, MMF 1547.5 |
In-vivo coronary artery of a swine |
54 to 88 in aortic arch 60 to 100 in right coronary artery |
FPI cavity with Hydrofluoric acid, SiO2 diaphragm |
- |
[52]83 |
|
SMF 1549.5 to 1550.5 1559.5 to 1560.5 |
Neck and ankle |
106 to 119 and 109 to 122 |
MZI, FBG |
7 and 5 on systolic |
[53]84 |
|
SMF, MMF |
tortuous vessels of a swine model in-vivo |
54 to 88 |
FPI, stretched core MMF, SiO2 diaphragm |
- |
[54]85 |
|
SMF 1549.5 to 1550.5
|
Right wrist |
110.7 supine 107.3 sitting 103.8 Standing |
MZI, FBG, SiO2, InGaAs |
3 supine 2.8 sitting 3.8 standing |
[55]86 |
|
SMF 1525 to 1575 |
Wrists |
123.6 average with no cover 119.2 average with cover |
FBG |
2 no covered 6 covered |
[56]87 |
|
SMF 1550 |
Wrist |
116.5/71.75 average
|
Erbium, Al diaphragm |
- |
[50]88 |
|
Plastic optical fiber (POF) 15543 and 1553 |
Left arm |
106.5/65 average |
FBG, glue NORLAND 78 |
5/3 |
[57]89 |
Table 6 – Multiparametric – vital signs, works on optical fiber sensors
|
Number of sensors |
Body temperature |
Respiration or Breathing rate |
Pulse or Heart rate |
Blood pressure |
Ref. |
|
3 |
× |
× |
× |
|
[6]8 |
|
2 |
|
× |
× |
|
[7]9 |
|
1 |
|
× |
× |
|
[35]18 |
|
1 |
× |
× |
|
|
[16]33 |
|
2 |
|
× |
× |
|
[36]51 |
|
2 |
× |
× |
× |
|
[22]79 |
|
1 |
|
|
× |
× |
[50]88 |
|
2 |
× |
× |
× |
|
[23]91 |
|
1 |
|
× |
× |
|
[34]92 |
|
1 |
|
× |
× |
|
[43]93 |
|
2 |
× |
× |
× |
|
[24]94 |
|
1 |
× |
× |
× |
|
[25]95 |
Table 6 only shows those works on fiber optic sensors that are capable of obtaining measurements of 2 or more vital signs; however, for ease of comparison, the data of most interest to us were included in the corresponding tables.
Tables of contents and abbreviations should be added to the manuscript. The abbreviations used throughout the text should be uniformized. For instance, in the introduction, the authors have consistently used the extended term "optical fiber". In section 2, suddenly they change for the abbreviation OF. Also, in many situations, the authors use semicolons, where it should be used periods.
The correction will become of all the abbreviations within the article
The general mechanisms of detection should be illustrated and explained briefly instead of constant repetition throughout the text. For instance, the major mechanism of detection for breathing and/or respiratory rate seems to be by sampling the human body's chest and abdomen movement. Afterwards, only the sensor characteristics should be discussed by the authors. This should be extended to the remaining sections if possible.
The literature on interferometers, mechanical induction, long-period fiber gratings, mechanically induced gratings and fiber Bragg gratings is expanded to support the article. Important data on vital signs are also added to complement the observations made.

Round 2
Reviewer 1 Report
The authors have provided extensive editing of the original manuscript. The revised version is suitable for publication.
Reviewer 2 Report
The authors made extensive modifications to address my concerns. This revised manuscript is suitable for publications.